# MULTITuDE: Large-Scale Multilingual Machine-Generated Text Detection Benchmark

**Dominik Macko, Robert Moro, Adaku Uchendu♠†, Jason Samuel Lucas†,**
**Michiharu Yamashita†, Matúš Pikuliak, Ivan Srba, Thai Le‡, Dongwon Lee†,**
**Jakub Simko, Maria Bielikova**

Kempelen Institute of Intelligent Technologies
{name.surname}@kinit.sk
♠ MIT Lincoln Laboratory
adaku.uchendu@ll.mit.edu
† The Pennsylvania State University, University Park, PA, USA
{jsl5710, michiharu, dongwon}@psu.edu
‡ University of Mississippi
thaile@olemiss.edu

## Abstract

There is a lack of research into capabilities of recent LLMs to generate convincing text in languages other than English and into performance of detectors of machine-generated text in multilingual settings. This is also reflected in the available benchmarks which lack authentic texts in languages other than English and predominantly cover older generators. To fill this gap, we introduce MULTITuDE[1], a novel benchmarking dataset for multilingual machine-generated text detection comprising of 74,081 authentic and machine-generated texts in 11 languages (ar, ca, cs, de, en, es, nl, pt, ru, uk, and zh) generated by 8 multilingual LLMs. Using this benchmark, we compare the performance of zero-shot (statistical and black-box) and fine-tuned detectors. Considering the multilinguality, we evaluate 1) how these detectors generalize to unseen languages (linguistically similar as well as dissimilar) and unseen LLMs and 2) whether the detectors improve their performance when trained on multiple languages.

## 1 Introduction

Machine text generation has significantly progressed in the past few months thanks to a new generation of large language models (LLMs). First, it was the arrival of ChatGPT and later GPT-4 that made available inexpensive generation of text in a range of languages to millions of people with ChatGPT becoming the fastest growing consumer application in history. Second, the introduction of LLaMA (Touvron et al., 2023b) opened new

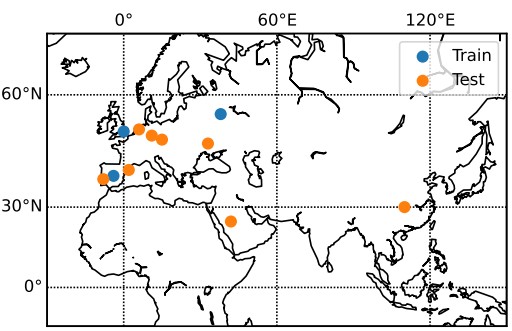

Figure 1: Train and Test languages from our dataset.

| Train | Test: **en** | Test: **non-en** | Difference |
|---|---|---|---|
| **en** | 0.9292 | 0.6903 | ↓ 25.7% |

Table 1: Average F1-score of detectors fine-tuned on English train split of MULTITuDE dataset, then evaluated on English test split vs. non-English test languages. The ↓ 25.7% drop in performance calls attention to the need for **accurate multilingual MGT detectors.**

possibilities for researchers and practitioners for inexpensive fine-tuning of LLMs, consequently ushering fine-tuned models like Alpaca (Taori et al., 2023) or Vicuna (Chiang et al., 2023) mimicking the capabilities of much larger (and more expensive) ones such as ChatGPT. The defining characteristic of this new generation of LLMs is not only the increased quality of text generation, but also their *multilinguality*.

Due to the potential for misuse of machine-generated text (MGT) for influence operations (Goldstein et al., 2023), disinformation (Buchanan et al., 2021), spam or unethical authorship (Crothers et al., 2022a), there has been a substantial amount of research on the task of

---

[1]The dataset is available at Zenodo upon request *for research purposes only*: https://zenodo.org/records/10013755. The source code is available at: https://github.com/kinit-sk/mgt-detection-benchmark.

*machine-generated text detection* (Jawahar et al., 2020; Stiff and Johansson, 2022; Uchendu et al., 2023a). Although GPT-3 was already capable of generating text in languages other than English and despite the availability of multilingual BLOOM (Scao et al., 2022), most of the prior works on this task have been (until very recently) still focusing on GPT-2 with English-only support or newer models like GPT-Neo (Gao et al., 2020), GPT-NeoX (Black et al., 2022) or GPT-J (Wang and Komatsuzaki, 2021) which were all trained on an English language only datasets.

Recently, first MGT datasets in languages other than English – AuTexTification for Spanish (Sarvazyan et al., 2023) and RuATD for Russian (Shamardina et al., 2022) – were made public for the detection task. There is also a dataset containing 5 languages (Chen et al., 2022), but this was obtained by the use of machine translation with human corrections, which renders it less useful for MGT detection benchmarking due to the potential noise. Thus, a dataset comprising authentic texts in multiple languages from a single domain (i.e., a text form, such as a news article or social media post) is still missing, hampering the comprehensive evaluation of detection methods in multilingual setting. At the same time, prior works have already shown that detectors fine-tuned on data in English fail to generalize to other languages (e.g. to German in case of Mitchell et al., 2023 showing a drop from 0.946 AUC ROC to 0.537), which is also confirmed by our results (see Table 1).

In this paper, we aim to address shortcomings of prior works and focus on the multilingual aspect of MGT detection task (a binary classification of a text to be human-written or machine-generated). Our main contributions are:

**(1)** We **evaluate the cross-language generalization** of fine-tuned detectors trained in monolingual vs. multilingual settings. More specifically, we evaluate how the detection methods fine-tuned to a specific language (monolingual) or to a set of training languages (multilingual) generalize to unseen languages. We observe strong influence of language family and script on generalization and clear benefits of multilingual fine-tuning.

**(2)** We provide a first comprehensive multilingual **benchmark of a range of state-of-the-art (SOTA) detection methods**, comparing the performance of *fine-tuned* detectors and their ability to

generalize to unseen LLMs to the performance of the *zero-shot* statistical detectors, such as Detect-GPT (Mitchell et al., 2023) or *black-box* methods, such as GPTZero.

**(3)** Finally, we introduce **a novel benchmarking dataset MULTITuDE** comprising of 74,081 texts (7,992 human-written and 66,089 machine-generated) in 11 languages (English, Spanish, Russian, Portuguese, Catalan, German, Dutch, Ukrainian, Czech, Arabic, and Chinese). The selected languages cover 4 different scripts and 5 language families (see Figure 1 for their geographic distribution). The included machine-generated texts were generated by 8 multilingual SOTA LLMs, namely GPT-3, ChatGPT, GPT-4, LLaMA-65B, Alpaca-LoRa-30B, Vicuna-13B, OPT-66B and OPT-IML-Max-1.3B covering various model sizes, architectures, and means of pre-training.

## 2 Related Work

**Large Language Models (LLMs).** They are language models with an unprecedented number of parameters trained on massive amounts of data, including models such as ChatGPT powered by GPT-3.5 or GPT-4 (OpenAI, 2023), OPT (Zhang et al., 2022), LLaMA (Touvron et al., 2023a), PaLM (Narang and Chowdhery, 2022), LaMDA (Collins and Ghahramani, 2021), BLOOM (Scao et al., 2022), Vicuna (Chiang et al., 2023), Alpaca (Taori et al., 2023), etc. The scale of LLMs has led to *emergent abilities*, observed only with these models, and solving of several non-trivial NLG and NLI (Natural Language Inference) tasks. Among the most impressive is the ability to generate authentic-looking human-like texts, nearly indistinguishable from human-written texts. Similarly impressive is the ability to generate coherent texts in languages other than English (Scao et al., 2022) as LLMs are mostly trained with over 50 languages. Because of these abilities, LLMs can be maliciously used, e.g. to generate misinformation (Shevlane et al., 2023). To combat LLM misuse, generated text detectors and benchmarks are required.

**Datasets.** Since transformer-based generative language models became ubiquitous in 2018, researchers have released datasets to address the new problem of *machine-generated text detection*. The most popular datasets are GPT-2[2] (Radford et al.,

---

[2]`https://github.com/openai/gpt-2-output-dataset`

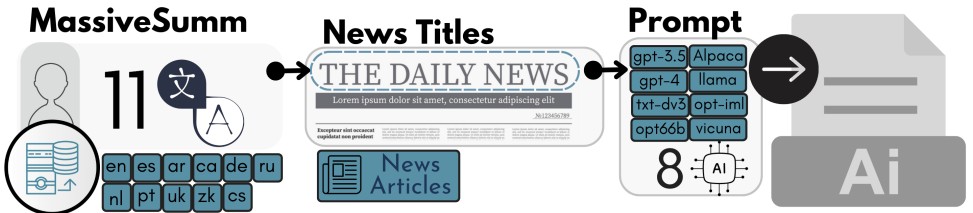

Figure 2: MULTITuDE generation framework.

2019) and GROVER[3] (Zellers et al., 2019) generated texts. However, as more Language Models (LM) got deployed, the need for more datasets from different LMs arose. Therefore Uchendu et al. (2020) released the first multi-generator (8 LMs) dataset in the news domain and Uchendu et al. (2021) released the first benchmark dataset (19 LMs) and environment for *machine-generated text detection*. Next, researchers released datasets for different domains - Academic papers (Liyanage et al., 2022; Rosati, 2022), News, Reddit posts & Recipes (Cutler et al., 2021), Amazon reviews (Adelani et al., 2020), multi-modal (i.e., images & texts) news articles (Tan et al., 2020), Tweets (Fagni et al., 2021), COVID-19 articles (Pagnoni et al., 2022), deepfake text in-the-wild (Pu et al., 2022a), gamification of MGT detection (Dugan et al., 2022), essays (Liu et al., 2023), prompt-generation (Anand et al., 2023; Peng et al., 2023), and multi-generator (27 LLMs) multi-domain (i.e., news, story, question, argument, scientific, etc.) data (Li et al., 2023).

However, the vast majority of these datasets are in English. A few researchers released non-English datasets in Russian (Shamardina et al., 2022), Chinese (Pu et al., 2022b), Iberian languages (Sarvazyan et al., 2023), and M4 which contains Russian, Chinese, Urdu, Indonesian, & Arabic languages (Wang et al., 2023). In light of the increasing number of multilingual LLMs, we generate the largest multilingual dataset for *machine-generated text detection* containing 11 languages.

**Detection Methods.** Prior works have shown that humans are already not capable of reliably distinguishing machine-generated from human-written text, with accuracy only slightly above random guessing (Uchendu et al., 2021) and even finding MGTs more trustworthy (Zellers et al., 2019). Thus, researchers have proposed a variety of automatic MGT detection methods. These include

stylometric-based, deep learning-based, statistics-based, and hybrid approaches (i.e., the ensemble of at least 2 approaches) (Uchendu et al., 2023a).

For stylometric-based detectors, researchers used linguistic features to capture the unique writing styles of the machine and human authors (Uchendu et al., 2020; Fröhling and Zubiaga, 2021; Kumarage et al., 2023). Due to the computational cost of extracting the linguistic features, researchers proposed a deep learning-based detector which involves fine-tuned and other variants of BERT (Zellers et al., 2019; Uchendu et al., 2021; Bakhtin et al., 2019; Liyanage et al., 2022; Rosati, 2022). However, deep learning models have a few limitations: (1) they are susceptible to adversarial perturbations (Gagiano et al., 2021; Crothers et al., 2022b; Wolff and Wolff, 2020) and (2) need a lot of labeled data to perform well. Thus, researchers proposed statistics-based techniques which are robust to adversarial perturbations and are unsupervised, requiring minimal data (Gehrmann et al., 2019; Mitchell et al., 2023; Gallé et al., 2021; Su et al., 2023). However, while these statistics-based detectors are more robust to perturbations than deep learning-based techniques, they still underperform deep learning-based models in terms of non-perturbed performance. Therefore, researchers combine statistics-based and deep learning-based techniques to gain adversarial robustness and high performance (Kushnareva et al., 2021; Liu et al., 2022; Uchendu et al., 2023b; Zhong et al., 2020).

## 3 Benchmark Dataset

As suitable multilingual human-machine pair dataset containing authentic non-English texts and machine texts generated by SOTA text-generation models is not available, we have put together a new dataset, called MULTITuDE (benchmark dataset for MULTIlingual machine Text DEtection). Its human segment comprises texts in 11 languages (authentic news articles) from the MassiveSumm dataset (Varab and Schluter, 2021) (see Figure 2).

---

[3] https://github.com/rowanz/grover/tree/master/generation_examples

| Language (Abbv) | Arabic (ar) | Catalan (ca) | Chinese (zh) | Czech (cs) | Dutch (nl) | English (en) | German (de) | Portuguese (pt) | Russian (ru) | Spanish (es) | Ukrainian (uk) | Total |
|---|---|---|---|---|---|---|---|---|---|---|---|---|
| Train | 0 | 0 | 0 | 0 | 0 | 26969 | 0 | 0 | 8970 | 8910 | 0 | 44786 |
| Test | 2673 | 2691 | 2683 | 2689 | 2695 | 2491 | 2685 | 2673 | 2671 | 2676 | 2668 | 29295 |

| Generator (Size) | alpaca-lora (30b) | gpt-3.5-turbo | gpt-4 | llama (65b) | opt (66b) | opt-iml-max (1.3b) | text-davinci-003 | vicuna (13b) | Total Machine | Total Human |
|---|---|---|---|---|---|---|---|---|---|---|---|
| Train | 5017 | 5023 | 5023 | 4998 | 4978 | 4956 | 5022 | 5013 | 40030 | 4756 |
| Test | 3273 | 3277 | 3277 | 3231 | 3251 | 3201 | 3275 | 3274 | 26059 | 3236 |

Table 2: Number of samples per language and per generator for train and test splits of the MULTITuDE dataset.

Titles of the selected articles have been used in the prompts for 8 language models to generate corresponding machine texts. The titles were split into train and test portion of the dataset, ensuring machine and human texts generated for the same title to be in the same split. The train split is used to fine-tune detectors in monolingual (a single training language) as well as multilingual (multiple training languages) manner; the test split of the dataset is used for evaluation of the detectors' performances. Details regarding text generation and pre-processing of both, human and generated, texts can be found in Appendix B.

**Language Selection.** We have deliberately selected major languages from three different language families as our training languages – English (Germanic), Spanish (Romance), and Russian (Slavic). To see whether linguistic similarity influences the transferability of the detection, we have selected two genealogically related test languages for all of them – Dutch and German for English, Czech and Ukrainian for Russian, Portuguese and Catalan for Spanish. On top of that, we have also generated test data for linguistically completely unrelated Arabic (Semitic) and Mandarin Chinese (Sino-Tibetan).

Most of the languages use Latin script. Russian is the only training language that uses a different script - Cyrillic. We have deliberately selected Czech (Latin) and Ukrainian (Cyrillic) as its Slavic neighbours to see how the script affects the results. Arabic and Chinese use their own scripts. Overall, our selection of languages is still biased towards Indo-European languages and Latin script.

The selected languages are just representatives to see the effect and answer our research questions, while avoiding waste of resources (including more languages than necessary). Using the published source code, the study can easily be extended to other languages.

**Generated Texts.** The summarized statistics of the MULTITuDE dataset are provided in Table 2. The dataset includes approximately 1000 human texts for each training language along with the corresponding MGTs from each generation model. For each test language, 300 human texts with the same amount of MGTs per model are included.

The linguistic analysis (see Appendix B.4) confirmed that all used text-generation models have been able to generate texts in the requested language in more than 95% cases (based on the Fast-Text language detection) except for LLaMA 65B (still reaching over 85%), failing mostly in Arabic and Chinese texts which it was not pre-trained on. The numbers of unique sentences and words per text is comparable to human texts and the results from the subsequent experiments show that none of the LLMs generated texts that are especially easy to detect. Nevertheless, some artifacts in the machine texts may still be present, since such a detailed analysis of the generated texts was not performed.

## 4 Detection Methods

For the purpose of this benchmark, the MGT detection methods are divided into three categories. The first category includes the *black-box detectors* – zero-shot methods available either through web interface or API, providing only small amount or no information about the underlying model or method used for detection, typically provided as commercial paid services. The second category includes *statistical detectors* – zero-shot methods, relying on distributional differences between generated and human-written text. The third category includes the *fine-tuned detectors* – language-model-based methods, which require fine-tuning of the models for the MGT detection task. See Appendix C for a complete list and more details on detection methods used in the benchmark.

**Black-Box Detectors.** We examine popular commercial black-box detectors, specifically ZeroGPT[4] and GPTZero[5]. Despite their wide use and support of non-English languages, the extent of their zero-shot multilingual and cross-lingual proficiency in detecting MGT remains unknown. The training methodologies, weight parameters, and the specific data used for these detectors remain undisclosed. We interacted with these detectors via a subscription-based API, enabling us to assess their performance on our multilingual dataset.

**Statistical Detectors.** We evaluate our dataset on all the current baseline and SOTA statistics-based detectors that had been previously shown to perform very well on English datasets (He et al., 2023; Mitchell et al., 2023), with all models (except Entropy) achieving over a 75% performance – Log-likelihood, Rank, Log-Rank, Entropy, GLTR Test-2, and DetectGPT. The main benefit of these techniques is that they require no training. Instead, they utilize the probability of each word in a piece of text to distinguish MGT from human-written texts. Typically, these statistics-based models use GPT-2 Medium to get the probability of the words, however, as we are evaluating with a multilingual dataset, we use mGPT, a multilingual GPT-based model. Further for DetectGPT, an additional model is used to generate perturbations to the original text, so we keep the default model for perturbation - T5 (Raffel et al., 2020).

**Fine-Tuned Detectors.** We have selected 7 most popular HuggingFace language models representing SOTA while taking multilinguality into account. In the experiments, these detector models have been fine-tuned on MGT detection task, taking various combinations of source languages and text generation models' output contained in the MULTITuDE dataset. For each of the three training languages separately (English, Spanish, Russian), for all training languages combined, and for English language with 3-times more train samples, we have fine-tuned these detector models for each generator separately and for all generators data combined. It resulted in 45 fine-tuned versions of each detector model, resulting in 315 fine-tuned detection methods in total. Details regarding fine-tuning process can be found in Appendix D.

[4] https://www.zerogpt.com
[5] https://gptzero.me/

## 5 Experiments and Results

We evaluate various aspects of multilingual capabilities of the existing SOTA MGT detection methods. Mainly, we focus on detection capabilities in English and non-English languages. However, we also analyze cross-lingual relations and generalization capabilities of detectors fine-tuned on a specific language and specific MGT generator data.

Firstly, in Table 3, we provide comparison of detectors' performance evaluated on the whole created multilingual MULTITuDE benchmark test data (i.e., the data balanced across 11 languages).

There are 315 fine-tuned detection methods, which is infeasible to show in a single table. For clarity, the table contains all evaluated black-box and statistical methods, but includes only the best-performing version of each base model of fine-tuned methods (i.e., only one fine-tuned version of XLM-RoBERTa is provided with information about language and generator LLM data used for training). Performance evaluation of all versions can be found in the associated GitHub repository. The results are ordered according to the achieved macro average F1-score (since the test data are highly imbalanced in terms of machine vs human classes). This metric is used in all experiments if not stated otherwise. In the table, we also show other standard performance metrics, such as weighted average of F1-score, Precision, Recall, Accuracy, and FPR (false positive rate) with FNR (false negative rate). These metrics are calculated based on a default classification threshold of 0.5. Such a threshold can be calibrated based on various aspects (such as minimizing FPR or maximizing Recall). Therefore, we also show AUC ROC (area under the curve of receiver operating characteristic), which is a threshold-independent metric calculated based on prediction probabilities rather than the predictions themselves. Unfortunately, due to missing prediction probabilities, it is available only for fine-tuned methods in our results. It must be noted that even when using optimal thresholds maximizing true positive rate and minimizing false positive rate, the key conclusions reported in this paper hold. We use the mentioned default threshold also when reporting results in the rest of this paper.

Based on the results, we can clearly see that fine-tuned methods outperform the others, when utilizing training data from all LLMs and all train languages (with two exceptions when fine-tuning on a single language performed better). We can

| Detector Model | Method Category | Train Lang. | Train LLM | Macro avg F1-score | Weighted avg F1-score | Weighted avg Precision | Weighted avg Recall | Accuracy | FPR | FNR | AUC ROC |
|---|---|---|---|---|---|---|---|---|---|---|---|
| MDeBERTa-v3-base* | F | all | all | 0.8480 | 0.9400 | 0.9403 | 0.9396 | 0.9396 | 0.2614 | 0.0354 | 0.9607 |
| XLM-RoBERTa-large* | F | all | all | 0.8240 | 0.9352 | 0.9357 | 0.9398 | 0.9398 | 0.4178 | 0.0158 | 0.9658 |
| BERT-base-multilingual-cased* | F | all | all | 0.7563 | 0.9073 | 0.9051 | 0.9104 | 0.9104 | 0.4781 | 0.0414 | 0.9188 |
| RoBERTa-large-OpenAI-detector | F | all | all | 0.7360 | 0.8933 | 0.8968 | 0.8904 | 0.8904 | 0.4308 | 0.0698 | 0.8645 |
| mGPT* | F | ru | all | 0.7219 | 0.8976 | 0.8941 | 0.9048 | 0.9048 | 0.5751 | 0.0356 | 0.8780 |
| GPT-2 Medium | F | all | all | 0.6646 | 0.8668 | 0.8682 | 0.8654 | 0.8654 | 0.5850 | 0.0787 | 0.7899 |
| ELECTRA-large | F | en | all | 0.5559 | 0.7952 | 0.8310 | 0.7684 | 0.7684 | 0.6530 | 0.1793 | 0.6053 |
| Entropy + RandomForest* | S | N/A | N/A | 0.4863 | 0.8335 | 0.8050 | 0.8729 | 0.8729 | 0.9756 | 0.0217 | N/A |
| Rank* | S | N/A | N/A | 0.4708 | 0.8375 | 0.7913 | 0.8895 | 0.8895 | 1.0000 | 0.0000 | N/A |
| DetectGPT* | S | N/A | N/A | 0.4708 | 0.8375 | 0.7913 | 0.8895 | 0.8895 | 1.0000 | 0.0000 | N/A |
| Entropy* | S | N/A | N/A | 0.4708 | 0.8375 | 0.7913 | 0.8895 | 0.8895 | 1.0000 | 0.0000 | N/A |
| Log-likelihood* | S | N/A | N/A | 0.4703 | 0.8368 | 0.7911 | 0.8880 | 0.8880 | 1.0000 | 0.0018 | N/A |
| Log-Rank* | S | N/A | N/A | 0.4702 | 0.8364 | 0.7911 | 0.8874 | 0.8874 | 1.0000 | 0.0025 | N/A |
| GLTR Test-2 (Rank)* | S | N/A | N/A | 0.4662 | 0.8282 | 0.7901 | 0.8707 | 0.8707 | 0.9991 | 0.0213 | N/A |
| ZeroGPT* | B | N/A | N/A | 0.4259 | 0.5559 | 0.8653 | 0.4744 | 0.4744 | 0.1681 | 0.5700 | N/A |
| GPTZero | B | N/A | N/A | 0.1605 | 0.1258 | 0.8636 | 0.1629 | 0.1629 | 0.0226 | 0.9383 | N/A |

Table 3: General performance of detection methods on the whole test split (all languages) of the MULTITuDE benchmark. Symbol * denotes detectors capable to handle multilingual text. Letters "B", "S" and "F" denote the category of the detector as black-box, statistical and fine-tuned respectively. Due to space limitations, we only report the best-performing version of each base model in case of fine-tuned detectors.

also notice that zero-shot methods cannot clearly distinguish between human and machine texts generated by newest LLMs. However, this is evaluated across data of all test languages combined; the results can differ among languages. In the following subsections, we thus report the results per individual test languages.

## 5.1 Zero-Shot Setting

We aim to answer the following research question: *How are zero-shot (statistical and black-box) detectors capable of detecting MGT in multiple languages?* The objective is to see how well these detectors can detect machine text generated by the newest LLMs and whether these detectors are able to detect MGT in non-English languages.

To answer this question, we run these detectors on the test split of the MULTITuDE dataset and analyze their per-language performances.

(1) **Statistical detectors cannot cope with multilingual data.** From Table 3, we observe that these models achieve about 47% F1 score. This suggests that these statistics-based models are unable to perform well with this multilingual constraint. Also, we observe that Rank, Entropy, and DetectGPT achieve the same performance. This is because all 3 models only predict one class, the machine class. The MGTBench[6] implementation of these methods, used in our experiments, uses a Logistic Regression classifier for binary predictions with default parameters. For the Entropy based method, we have also used a Random Forest classifier with hyperparameters optimized using Randomized Grid

Search with 5-fold cross-validation and 1k of iterations (details regarding the optimized hyperparameters can be found in Appendix D.), achieving a slightly higher performance. Notably, we can see that such a model is predicting also a human class, although negligibly, meaning the method actually works. Finally, the low performance of these previously high-performing statistical models suggests the non-trivial nature of evaluating on a multilingual dataset.

(2) **Transferability to non-English languages cannot be properly evaluated.** It is due to the overall low performance (e.g., predicting a single class only) of the statistical and black-box detectors (even on English, as previously mentioned). Per-language results show that black-box detectors outperformed statistical detectors on English data, but their performance on other languages is the same or even worse (see Table 10 in Appendix E).

## 5.2 Monolingual Generalization

In this experiment, we aim to answer the following research question: *Do detectors fine-tuned in monolingual settings generalize to other languages?* Meaning, for example, will a detector fine-tuned on English data only perform well on Spanish? Is there a relation between how close languages are and how well the detectors generalize?

To answer this question, we use various versions of detectors, fine-tuned on individual language data. To better show language dependencies, we perform this experiment per each generator data separately. For example, the XLM-RoBERTa model is fine-tuned on GPT-4 machine data (plus human data)

---

[6] https://github.com/xinleihe/MGTBench

| Train | Test Language [mean (±confidence interval)] | | | | | | | | | | |
|---|---|---|---|---|---|---|---|---|---|---|---|
| Language | ar | ca | cs | de | en | es | nl | pt | ru | uk | zh |
| en | 0.5448 | 0.7335 | 0.6793 | 0.8104 | 0.9292 | 0.7018 | 0.7508 | 0.7362 | 0.7148 | 0.6746 | 0.5580 |
| | (±0.07) | (±0.07) | (±0.06) | (±0.04) | (±0.02) | (±0.05) | (±0.07) | (±0.05) | (±0.05) | (±0.05) | (±0.05) |
| es | 0.7857 | 0.8747 | 0.8016 | 0.8812 | 0.7322 | 0.9314 | 0.8143 | 0.8944 | 0.8375 | 0.8299 | 0.7216 |
| | (±0.05) | (±0.03) | (±0.07) | (±0.03) | (±0.07) | (±0.02) | (±0.06) | (±0.03) | (±0.04) | (±0.05) | (±0.06) |
| ru | 0.8487 | 0.6532 | 0.7924 | 0.7591 | 0.5760 | 0.6884 | 0.6915 | 0.6626 | 0.9522 | **0.9387** | 0.7294 |
| | (±0.05) | (±0.07) | (±0.08) | (±0.06) | (±0.09) | (±0.07) | (±0.07) | (±0.07) | (±0.01) | (±0.02) | (±0.06) |
| all | **0.8537** | **0.8977** | **0.8604** | **0.9073** | **0.9420** | **0.9372** | **0.8808** | **0.9253** | **0.9560** | 0.9374 | **0.7659** |
| | (±0.04) | (±0.03) | (±0.07) | (±0.02) | (±0.02) | (±0.02) | (±0.04) | (±0.02) | (±0.01) | (±0.02) | (±0.04) |
| en3 | 0.5605 | 0.7484 | 0.7289 | 0.8244 | 0.9392 | 0.7156 | 0.7778 | 0.7508 | 0.7092 | 0.7118 | 0.6160 |
| | (±0.09) | (±0.06) | (±0.07) | (±0.04) | (±0.03) | (±0.04) | (±0.07) | (±0.05) | (±0.06) | (±0.06) | (±0.05) |

Table 4: Performance for the test languages based on various train language combinations. It shows the mean of all trained detectors with multilingual base models along with 95% confidence interval error bounds. The reported score is macro average F1-score.

from train split for English, and evaluated on GPT-4 machine data (plus human data) from test split for all languages separately (English, Spanish, etc.).

Table 4 shows the aggregated performance across all generators and all multilingual detectors (i.e., detectors having a multilingual base LM). We only use multilingual detectors here because they have the best performance, as the cross-lingual generalization capability of English-only models is worse (see Table 3). For each test language, we test whether the differences between train languages observed in Table 4 are statistically significant. To do this, we conduct repeated measures ANOVA tests for each test language: we use macro F1-score for a given test language as a dependent variable, the combinations of detectors and text generators as "subjects" and train language as an independent within-subjects variable. For all 11 test languages, the observed differences are statistically significant ($p < 0.05$). We further conduct post hoc pairwise tests between pairs of train languages per each test language for a more in depth analysis. We also show how the performance for individual languages correlate in Table 5. For completeness, we also provide full results in Appendix E in Tables 11–13 for English, Spanish and Russian training language respectively.

There are several observations that can be made based on our results. (1) The results confirm that the **monolingually fine-tuned detectors are able to generalize to other languages**, although with some performance degradation. There are significant differences of performance achieved for individual test languages (ranging from 0.54 to 0.96).

(2) **Linguistic similarity matters.** The results indicate that the similarity between languages plays

a role in how they would generalize between each other. Spanish dominates both Catalan and Portuguese, similarly Russian works really well for Ukrainian. The correlations also clearly show that the performance of similar languages correlate with each other. Czech is the one exception from this trend, but it might be caused by the fact that it is both Slavic (more similar to Russian), but it also uses the Latin script (making it more similar to other Latin-using Indo-European languages).

(3) **English is an outlier language.** Overall it has low (but statistically significant) correlation with other related languages and it is the only language that has negative correlation with any other language. It is outperformed by Spanish in most cases, even for the languages from its own language family; observed differences in performance between using Spanish or English as a training language are statistically significant for both German and Dutch. At the same time, detectors trained on other training languages (Spanish and Russian) have unusually weak performance for English. We hypothesize that this is caused by the fact that English is often the most common language in the pre-training data for both the generators and the detectors, which might lead to different behavior (regarding cross-lingual capability) for this particular language (e.g., the perplexity might be lower).

(4) **Languages with Non-Latin scripts are correlated.** Even though Russian is completely unrelated to Arabic or Chinese, it has the best performance as a training language (although the differences in performance when using Russian or Spanish as a training language are not statistically significant). The Non-Latin script languages seem to correlate well with each other. This might indi-

| | Germanic Languages | | | Romance Languages | | | Slavic Languages | | | Others | |
|---|---|---|---|---|---|---|---|---|---|---|---|
| **Languages** | **en** | **de** | **nl** | **es** | **pt** | **ca** | **cs** | **ru** | **uk** | **ar** | **zh** |
| en | 1.0000 | 0.5420 | 0.5551 | 0.3794 | 0.4960 | 0.4237 | -0.16 (n.s.) | -0.3235 | -0.4988 | -0.2672 | -0.00 (n.s.) |
| de | 0.5420 | 1.0000 | 0.6006 | 0.7657 | 0.8022 | 0.6491 | 0.2176 | 0.20 (n.s.) | 0.08 (n.s.) | 0.20 (n.s.) | 0.17 (n.s.) |
| nl | 0.5551 | 0.6006 | 1.0000 | 0.5585 | 0.6905 | 0.8342 | 0.06 (n.s.) | 0.2403 | 0.05 (n.s.) | 0.3516 | 0.4694 |
| es | 0.3794 | 0.7657 | 0.5585 | 1.0000 | 0.9317 | 0.7331 | 0.16 (n.s.) | 0.18 (n.s.) | 0.12 (n.s.) | 0.2989 | 0.2015 |
| pt | 0.4960 | 0.8022 | 0.6905 | 0.9317 | 1.0000 | 0.8251 | 0.09 (n.s.) | 0.13 (n.s.) | 0.05 (n.s.) | 0.2483 | 0.19 (n.s.) |
| ca | 0.4237 | 0.6491 | 0.8342 | 0.7331 | 0.8251 | 1.0000 | 0.15 (n.s.) | 0.2103 | 0.08 (n.s.) | 0.3345 | 0.3160 |
| cs | -0.16 (n.s.) | 0.2176 | 0.06 (n.s.) | 0.16 (n.s.) | 0.09 (n.s.) | 0.15 (n.s.) | 1.0000 | 0.3690 | 0.4489 | 0.4264 | 0.4500 |
| ru | -0.3235 | 0.20 (n.s.) | 0.2403 | 0.18 (n.s.) | 0.13 (n.s.) | 0.2103 | 0.3690 | 1.0000 | 0.8606 | 0.7378 | 0.4463 |
| uk | -0.4988 | 0.08 (n.s.) | 0.05 (n.s.) | 0.12 (n.s.) | 0.05 (n.s.) | 0.08 (n.s.) | 0.4489 | 0.8606 | 1.0000 | 0.7398 | 0.4664 |
| ar | -0.2672 | 0.20 (n.s.) | 0.3516 | 0.2989 | 0.2483 | 0.3345 | 0.4264 | 0.7378 | 0.7398 | 1.0000 | 0.7249 |
| zh | -0.00 (n.s.) | 0.17 (n.s.) | 0.4694 | 0.2015 | 0.19 (n.s.) | 0.3160 | 0.4500 | 0.4463 | 0.4664 | 0.7249 | 1.0000 |
| | Latin Script | | | | | | Non-Latin Script | | | | |

Table 5: The correlations between the macro average F1-score performance of the test languages calculated based on the results from multilingual detectors (i.e., having a multilingual base LM). The results that are not statistically significant are marked by (n.s.).

cate that the models behave differently for the Latin script (which is, again, by far the most common) than for other scripts.

## 5.3 Multilingual Generalization

In this experiment, we aim to answer the following research question: *Do detectors fine-tuned in multilingual settings generalize better to unseen languages than monolingually fine-tuned ones?* The objective is to see whether it is beneficial to train detectors on multilingual rather than on monolingual data in regard to transferability to other languages.

For the purpose of this experiment, we fine-tune the detectors by using the train samples of all three languages combined. The train set consists of 1k MGTs and 1k human texts for each train language (this train set is denoted as *all* in the results). The evaluation is also done per each LLM separately. In order to see whether the performance is not strictly based on a higher number of train samples, we also fine-tune the detectors with a comparable amount of English-only data, i.e., also approximately 6k train samples (this train set is denoted *en3*). The mean results are provided in bottom two rows of Table 4, analogously to the previous experiment. For completeness, the full results of each detector per each LLM-generated data are provided in Appendix E in Tables 14 and 15.

**The multilingually fine-tuned detectors perform better on unseen languages than the monolingually fine-tuned ones**. The observed differences in performance between using all three train languages (*all*) and all other train setups are statistically significant in case of Czech, German, Dutch and Portuguese. For all other test languages, the multilingually fine-tuned detectors also perform better (with the sole exception of the Ukrainian language where the detectors trained on Russian

slightly outperform the ones trained on *all*), but the differences to the best monolingually fine-tuned detectors are not statistically significant. The reason for a better performance of the multilingually fine-tuned detectors may be a higher amount of training samples. Indeed, when we look at the results of detectors fine-tuned with the *en3* train set, they achieve a slightly (but mostly not statistically significantly) better performance in almost all cases compared to the original English fine-tuned detectors (performing almost the same as multilingually fine-tuned detectors on English). However, the generalization to other languages is still significantly better in multilingually fine-tuned detectors (*en3* having a minimum at 0.56 for Arabic vs. *all* having a minimum at 0.77 for Chinese). Thus, regarding transferability to other languages, the detectors fine-tuned in multilingual manner seem stronger (for detectors with multilingual base models as well as for the ones with monolingual base models; see Appendix E for the latter).

## 5.4 Cross-Generator Generalization

We also aim to answer the research question: *How do fine-tuned detectors trained on data from a single LLM perform in detecting MGT by different LLMs?* Analogously to cross-lingual evaluation in Section 5.2, the objective of this experiment is to scrutinize the cross-generalizability among distinct LLMs by each individual detector fine-tuned on data from a single LLM.

Table 6 shows the correlation in the performance of each individual LLM. Comprehensive results (i.e., all the macro F1-scores, mean, and standard deviation separated per each language) are provided in Appendix E in Tables 16–20.

**LLM similarity matters.** We discern two distinct groups, namely, **Group 1**: text-davinci-003,

| | text-davinci-003 | gpt-3.5-turbo | gpt-4 | alpaca-lora-30b | vicuna-13b | llama-65b | opt-66b | opt-iml-max-1.3b |
|---|---|---|---|---|---|---|---|---|
| **text-davinci-003** | 1.0000 | 0.9585 | 0.9005 | 0.9357 | 0.9381 | -0.4537 | -0.3444 | -0.3574 |
| **gpt-3.5-turbo** | 0.9585 | 1.0000 | 0.9712 | 0.8562 | 0.9056 | -0.5131 | -0.4805 | -0.4872 |
| **gpt-4** | 0.9005 | 0.9712 | 1.0000 | 0.7781 | 0.8786 | -0.4868 | -0.4779 | -0.5218 |
| **alpaca-lora-30b** | 0.9357 | 0.8562 | 0.7781 | 1.0000 | 0.9268 | -0.2870 | -0.1261 | -0.1226 |
| **vicuna-13b** | 0.9381 | 0.9056 | 0.8786 | 0.9268 | 1.0000 | -0.2221 | -0.1273 | -0.1632 |
| **llama-65b** | -0.4537 | -0.5131 | -0.4868 | -0.2870 | -0.2221 | 1.0000 | 0.7721 | 0.6990 |
| **opt-66b** | -0.3444 | -0.4805 | -0.4779 | -0.1261 | -0.1273 | 0.7721 | 1.0000 | 0.9011 |
| **opt-iml-max-1.3b** | -0.3574 | -0.4872 | -0.5218 | -0.1226 | -0.1632 | 0.6990 | 0.9011 | 1.0000 |

Table 6: The correlations between the macro average F1-score performance of the cross-generator. All the presented results are statistically significant.

gpt-3.5-turbo, gpt-4, alpaca-lora-30b, and vicuna-13b, and **Group 2**: llama-65b, opt-66b, and opt-iml-max-1.3b. These groups have positive and negative correlations with each other. Group 1 primarily consists of models developed by OpenAI such as text-davinci-003, gpt-3.5-turbo, and gpt-4. Alpaca is a derivative fine-tuned model from a LLaMA 7B model on 52K instruction-following demonstrations generated from text-davinci-003 (Taori et al., 2023), while Vicuna is also fine-tuned with 70K user-shared ChatGPT conversations and achieves 92 % ChatGPT (i.e., gpt-3.5-turbo) quality (Chiang et al., 2023). Hence, we consider these as OpenAI-based models. On the other hand, Group 2 encompasses models developed and released by Meta AI, hence recognized as Meta-based models. The LLM architectures within each group are similar, which may cause the similar fine-tuned performance on the dataset and this observed phenomenon.

## 6 Discussion

**Multilingual fine-tuned detectors perform the best.** Fine-tuning multilingual LMs is the best approach based on our results, outperforming both English LMs and statistical methods. They have better ability to generalize to other unseen languages unmatched by the other methods. Still, their performance on the MULTITuDE dataset is far from perfect and the best model (MDeBERTa-v3-base) achieved Macro F1 score of 0.85. As such, they can not be used to reliably detect MGTs.

**Linguistic similarity matters.** Our results show that the linguistic similarity between the languages influence how well they generalize to each other and how much the performance for the languages correlate. The typology of the languages, but also the script they use are important. Multilingual MGT detectors should be trained and tested with a diverse set of languages to ensure the inclusivity of their performance. Yet, the practical development is often hindered by the fact that different LMs (used both as generators and detectors) support different sets of languages to different extent, making it hard to create one-model-fits-all solutions.

**English is an inappropriate default.** As mentioned previously, the performance on the English language is an outlier and the models do not generalize to other languages that well from this language. English is often used as the *de facto* default language for many NLP use cases, including using it as a source language for crosslingual learning. This should be reconsidered for multilingual MGT detection.

## 7 Conclusion

In the paper, we provide the first comprehensive benchmark of black-box, statistical and fine-tuned machine-generated text detection methods in multilingual settings using our novel MULTITuDE dataset, covering 11 languages and 8 SOTA LLMs. Our results show that most currently available black-box methods do not work in multilingual settings and that the statistical approaches lag behind the fine-tuned ones. We also show that fine-tuning models in a multilingual manner (i.e., train data in multiple languages) results in better performance of detectors for unseen languages compared to monolingual fine-tuning. The generalization is strongly affected by language script as well as language family branches of the train and test languages. Also, English seems a particularly inappropriate choice of a training language if one aims for generalization of machine-generated text detection to non-English languages. This further emphasizes the importance of creating multilingual benchmarks for machine-generated text detection such as MULTITuDE. As a future work, we plan to extend it with a more diverse set of languages (in terms of scripts and language families) and with texts from other domains, especially social media.

## Limitations

**Language Selection.**  Our work is limited by the final selection of 3 training and 11 testing languages. This already allowed us to discover that there are interesting linguistic properties in the detector methods, but based on our work we still can not tell how they would behave with all the other languages. Non-European languages especially are still a blind spot in our evaluation.

**Limited Amount of Training Data.**  Another limitation, closely related to the previous one is the fact, that the amount of data we use for benchmarking is limited. Apart from using different languages, the data could also be expanded by different domains, writing styles, etc. The amount of training data we use is also limited (several thousand samples), and simply extending the existing data could also yield additional improvements.

**Limited Selection of Generative Language Models.**  In the end, we have selected and experimented with 8 generative language models, which are capable of generating multilingual content. It is hard to ascertain how generalizable the results are for all the other language models that are being or will be developed in future with different training data and different training regimes.

## Ethics Statement

As a part of this paper, we introduce the MULTITuDE dataset consisting of human-written and machine-generated texts. The human-written texts are news articles collected in the MassiveSumm dataset (Varab and Schluter, 2021). The MassiveSumm dataset does not specify a license under which they publish the data as its public version only contains a list of URLs and a software package for their downloading and processing. Thus, we can assume that the news articles are protected by copyright, which, however, allows their use for non-commercial research such as our work. Although most of the LLMs we used were hosted at our premises, we also used OpenAI API. As a part of the prompts, we were sending headlines of the news articles to the API; these, however, are not used by the OpenAI to train or improve their models (which would constitute a commercial use) and they are retained for a maximum duration of 30 days, after which they are deleted.[7]

Regarding the used LLMs, we made sure to follow their terms of use as well. LLaMA models (and their variants Alpaca and Vicuna) are licensed for non-commercial use only.[8] Additionally, outputs of OpenAI services cannot be used to "develop models that compete with OpenAI."[9] Respecting these limitations, we publish the MULTITuDE dataset containing both the human-written texts with attribution (original source) and the machine-generated texts only for *non-commercial research purposes*.

**Intended Use.**  The collected dataset is primarily intended to be used for research of multilingual machine-generated text detection. We used it for binary classification, but it could also be employed for multi-class classification (i.e., *authorship attribution* as defined in Uchendu et al., 2021, 2023a). We also publish the code for analysis and reproduction of our results including the training (fine-tuning) of the detection methods. These are also intended for research purposes only. They are not intended (in their current form) to be used in actual deployment where they would be automatically classifying the texts as human-written or machine-generated.

**Failure Modes.**  As already noted in limitations, the fine-tuned detectors, while showing promising performance in our experiments, might fail when used on unseen languages, texts from different domains or writing styles. Additionally, they can fail to generalize to other unknown LLMs, decoding strategies or obfuscation efforts. The potential harms are not only from false negatives (i.e., failing to detect machine-generated texts), but also (and potentially even more so) from false positives (i.e., falsely flagging a text as being machine-generated while it was in fact human-written). It is also worth noting that there are many non-malicious uses of machine-generated texts (e.g., proofing, translation, etc.), which needs to be considered before any use of the detection methods trained on our collected dataset for purposes beyond research.

**Biases.**  Although we selected languages from different language families and with different scripts (see Section 3), the dataset is still biased towards Indo-European languages and Latin script. Because of the nature of the training data which consists of news articles written in a standardized form

---

[7] https://openai.com/policies/api-data-usage-policies

[8] https://github.com/facebookresearch/llama/blob/main/MODEL_CARD.md

[9] https://openai.com/policies/terms-of-use

of each included language, detectors fine-tuned on the dataset might be biased with respect to use of dialects, slang or code-switching which could potentially harm individuals from some ethnic groups or social origins.

**Misuse Potential.** We believe that there is only a limited possibility of misuse of our dataset. First, the dataset is published for research purposes only. Second, the machine-generated texts, although inauthentic and most likely false, should not cover any sensitive topics. Also, the used prompts to LLMs were to generate news articles given a headline; we did not prompt the LLMs to intentionally generate disinformation. So their potential harm and impact in case of misuse is limited.

**Collecting Data from Users.** The collection and processing of the dataset did not include any crowd workers or any other annotators. We do not intentionally collect or store any personal data as a part of this research. Some personal data (e.g., names) might be generated by the LLMs, but we can assume these to be mostly public figures that could have appeared in the training data of LLMs.

**Potential Harm to Vulnerable Populations.** To the best of our knowledge, the dataset does not cover any sensitive topics beyond what is normally covered in the news. As already noted in *Biases*, the dataset does not include texts in different writing styles, dialects or slang which can be used by marginalized populations and the detectors fine-tuned on the dataset could thus fail in such cases.

## Acknowledgements

This work was partially supported by the *VIGI-LANT - Vital IntelliGence to Investigate ILlegAl DisiNformaTion*, a project funded by the European Union under the Horizon Europe, GA No. 101073921, by *vera.ai - VERification Assisted by Artificial Intelligence*, a project funded by European Union under the Horizon Europe, GA No. 101070093, and by NSF awards #1820609, #1934782, #2114824, and #2131144.

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

# A  Computational Resources

For the purpose of data pre-processing and results analysis, we have used Google Colab[10] without GPU acceleration, leaving thus only small carbon footprint on the environment. For the machine text generation, we have used multiple resources. For the OpenAI LLMs, we have used OpenAI API, requiring no GPU acceleration on our side. For LLaMA 65B, Alpaca LoRa 30B, and OPT 66B models, we have used $1\times$ A100 40GB GPU ($4\times$ A100 for >60B models), with a cumulative run-time of approximately 32 GPU-days. For Vicuna 13B and OPT-IML 1.3B models, we have used $1\times$ NVIDIA GeForce RTX 3090 24GB GPU, with a cumulative run-time of approximately 8.5 GPU-days. Regarding detectors fine-tuning, we have used $1\times$ NVIDIA GeForce RTX 3090 24GB GPU, with a cumulative run-time of approximately 11.3 GPU-days. For running black-box detectors, we have also used Google Colab without GPU acceleration. For running statistical methods requiring a base LLM (i.e., mGPT), we have used $1 \times$ A100 40GB GPU on an a2-highgpu-1g machine type.

# B  Data Pre-processing

For MULTITuDE dataset creation, we have used human texts from the original articles included in the MassiveSumm[11] (Varab and Schluter, 2021) dataset. We have used on-request author-provided processed data as well as CommonCrawl based links published in the GitHub repository. Both sources result in per-language datasets (split into files according to languages).

## B.1  Human-Text Pre-processing

We have selected 3 languages for training (detectors fine-tuning) and 8 more for testing (11 languages in total). For each of the selected languages, we have taken the first 50,000 samples (if available for that language) from each data source. It resulted up to 100,000 samples per language.

The texts of these samples were stripped, meaning that white-space characters from beginning and end of texts are removed. Out of these samples, we have dropped samples with missing values and duplicated samples. Then, we have dropped samples that contained texts with 5 or less words (where the "word" is represented as a white-space delimited substring).

---

[10] https://colab.research.google.com/
[11] https://github.com/danielvarab/massive-summ

| Language | Original | N/A & Duplicates Removed | Min Textsize Applied | Language Checked |
|---|---|---|---|---|
| ar | 73990 | 67351 | 67267 | 56140 |
| ca | 804 | 804 | 804 | 569 |
| cs | 97136 | 86507 | 84797 | 59039 |
| de | 69300 | 69127 | 69099 | 54914 |
| en | 42050 | 35023 | 34611 | 8812 |
| es | 99594 | 92484 | 91569 | 38254 |
| nl | 1797 | 1797 | 1797 | 1489 |
| pt | 95444 | 90209 | 90020 | 38857 |
| ru | 95797 | 88743 | 88304 | 51938 |
| uk | 100000 | 92868 | 90924 | 39844 |
| zh | 92698 | 81812 | 56678 | 8875 |
| **All** | 768610 | 706725 | 675870 | 358731 |

Table 7: Overview of the amount of language samples from the MassiveSumm dataset remaining after each pre-processing step.

The MassiveSumm per-language datasets contained texts in other languages, meaning that some texts were in different languages than the intended language according to the file (identified in the filename). Therefore, we have performed language detection of samples and dropped those that did not match. For language detection, we have used a combination of FastText[12] (Joulin et al., 2017) and polyglot[13] tools. We have taken the language predictions into account only if the probability score ("confidence") was at least 0.9. We have performed such detection separately using the title and the first sentence of the text for a given sample, resulting in 4 predictions. Out of these a majority voting was used to give a final detection result.

Table 7 provides amounts of texts per language available after each of the above mentioned pre-processing step.

After pre-processing, we have pseudo-randomly sub-sampled the English data to 3300 samples, Spanish and Russian to 1300 samples, and others to 300 samples. 300 samples of each language are then used for test split, while the remaining ones are in train split of the dataset.

The selected numbers of samples are based on our preliminary study using the existing datasets: TuringBench English data (Uchendu et al., 2021), AuTexTification Spanish data (Sarvazyan et al., 2023), and RuATD Russian data (Shamardina et al., 2022). We have extensively experimented and chose a minimal number of samples needed to fine-tune the detectors and properly evaluate them. Specifically, we have compared the performances using the selected smaller amount of samples (500, 600, 1000, 1500 human samples where available)

and all samples available (5000 for English, 1150 for Spanish, 2450 for Russian). These experiments resulted in 1k human samples and 1k samples per text-generator required for training, with a negligible drop in the performance of fine-tuned detectors (i.e., within 5 %). For testing, 300 texts per class shown to be enough, giving the same detector performance values as using all the available samples. In addition, since we are experimenting with smaller number of samples in train sets, we have provided 3k human texts for English train data to ensure that the performance effect is not simply due to a higher number of train samples in case of multilingually fine-tuned detectors.

## B.2 Machine Text Generation

Titles (i.e., headlines) of the human-text samples have been used in prompts to generate corresponding machine texts by multiple large language models. For instruction-following models, an instruction-based prompt has been used (a universal prompt in English instructing to generate text in a target language, corresponding to the title in the target language), pure title in the others. The instruction-based prompt was used in the form as follows (where language_name and headline are variables inserting strings specifying language and title of an individual text sample):

> You are a multilingual journalist.\n\nTask: Write a news article in {language_name} for the following headline: "{headline}". Leave out the instructions, return just the text of the article.\n\nOutput:

Settings for the text generation include minimal length of generated text set to 200 tokens, maximal length of generated text set to 512 tokens, number of return sequences set to 1, sampling activated with beams number of 1, top_k of 50, and top_p of 0.95. For models available via OpenAI API, we have set only maximal number of tokens to 512 and top_p to 0.95.

## B.3 Generated-Text Pre-processing

Pre-processing of the generated texts included text stripping (i.e., white-space characters from beginning and end of texts are removed), removal of prompts from the generated text (both title and instruction-based prompt), removal of unfinished sentence from the end of the text (if more sentences are present). In order to achieve similar text

[12] https://github.com/facebookresearch/fastText/
[13] https://github.com/aboSamoor/polyglot

| Generator | Language Match | Empty Text | Short Text | WC mean | WC std | US mean | US std | UW mean | UW std |
|---|---|---|---|---|---|---|---|---|---|
| text-davinci-003 | 100.00 | 0 | 3 | 136.57 | 76.32 | 1.00 | 0.00 | 0.67 | 0.14 |
| gpt-3.5-turbo | 99.98 | 0 | 0 | 152.22 | 76.85 | 1.00 | 0.00 | 0.65 | 0.13 |
| gpt-4 | 100.00 | 0 | 0 | 208.86 | 121.91 | 1.00 | 0.00 | 0.64 | 0.13 |
| alpaca-lora-30b | 98.99 | 0 | 10 | 115.78 | 59.27 | 1.00 | 0.02 | 0.68 | 0.12 |
| vicuna-13b | 97.11 | 0 | 12 | 155.40 | 80.64 | 1.00 | 0.02 | 0.62 | 0.14 |
| llama-65b | 85.71 | 0 | 71 | 150.79 | 87.76 | 0.98 | 0.07 | 0.59 | 0.15 |
| opt-66b | 96.47 | 0 | 71 | 152.26 | 103.62 | 1.00 | 0.03 | 0.67 | 0.15 |
| opt-iml-max-1.3b | 95.20 | 7 | 143 | 154.04 | 112.30 | 0.99 | 0.08 | 0.61 | 0.19 |
| human | 99.89 | 0 | 0 | 136.82 | 78.16 | 1.00 | 0.02 | 0.69 | 0.12 |

Table 8: Statistics of the machine-generated texts. Mean and standard deviation of ratios are provided, where WC refers to the word count, US refers to the unique sentences, and UW refers to the unique words.

lengths distribution between human and machine texts, each machine text is shortened if the corresponding human text (title of which was used in the prompt) is shorter. Similarly, the human texts are shortened to a mean value of lengths of the corresponding machine texts. Shortening occurs only if the difference between the corresponding machine and human texts lengths is greater than 5 words and if more than one sentence is present. Shortening is performed by removal of the last sentence from the longer text until the condition is met. Such texts are then processed by the FastText full-text language detection, and language mismatch is analyzed (see the next subsection).

After the initial analysis of the generated texts, we have noticed multiple prompts were duplicated. In order to preserve consistency, we have moved all texts generated for the duplicated prompts to the same (train) split of the dataset. The intuition is to avoid having the texts generated for the same prompt in both splits. We have then dropped generated samples with 5 or less words and dropped text duplicates, ensuring no text sample has multiple labels. Fortunately, even after occurrence of duplicated texts and their removal from the final dataset, the numbers of samples are not significantly reduced. The smallest number in the train split is the number of human Spanish samples having 937 unique texts (out of intended 1000). The smallest number in the test split is the number of llama-65b English samples having 275 unique texts (out of intended 300). Thus, the numbers of samples for each text-generation language model and for each language are still well balanced.

### B.4 Linguistic Analysis of the Generated Text

The statistics of the analyzed generated texts per language model are provided in Table 8. For Chi-nese word count, polyglot tool was utilized. For other languages, white-space separated substrings are counted. The *Empty Text* column contains number of samples with no new text generated (i.e., returned only the provided prompt or no text at all). The *Short Text* column represents the number of generated texts with 5 or less words. As the table indicates, the llama-65b model performed worst in generating texts in multiple languages. But it still had more than 85% accuracy regarding language match (i.e., the language of the generated text is the same as the one queried by the prompt). We must also take into account the fact that FastText language identification is not error-free (i.e., misclassification can occur). As expected, deeper analysis revealed that llama-65b model have missed mostly Chinese and Arabic languages (202 and 140 mismatched samples, respectively), since these were not used in the model training.

## C Description of Detection Methods

Table 9 shows descriptions of all detection methods used in the benchmark. The base models for the detection methods have been carefully selected with respect to the state-of-the-art and the limited experimentation resources. We have primarily used multilingual base models (XLM-RoBERTa, BERT-multilingual, mGPT, and MDeBERTa) that are publicly available and belong to the SOTA multilingual pre-trained models used for a wide range of downstream tasks. Besides these, we have also used English-only pretrained models that have been commonly used as detectors in previous studies (Uchendu et al., 2021; Zhong et al., 2020). We used these to see how they would perform on non-English language datasets. In this group, there were RoBERTa-large-OpenAI-detector, GPT2, and ELECTRA.

| Detector | Category | Description |
|---|---|---|
| ZeroGPT | Black-box | ZeroGPT service uses a series of complex and deep algorithms to analyze the text, presented with an accuracy rate of text detection up to 98%, claiming to detect AI text output in all the available languages. https://www.zerogpt.com |
| GPTZero | Black-box | GPTZero model can detect AI-generated and human-written text across the sentence, paragraph, and document levels. Training on a mixed corpus of AI and human English writings, it can accurately classify 85% of AI and 99% of human texts using a 0.65 threshold. To reduce false positives, a 0.65 or higher threshold is advised. https://gptzero.me/ |
| Log-likelihood (Solaiman et al., 2019) | Statistical | Given a text, this method calculates the average word log probability of each word. The log probability is extracted from a language model (i.e., mGPT (Shliazhko et al., 2022)). |
| Rank (Gehrmann et al., 2019) | Statistical | Similar to Log-likelihood, given each word in a text, using the context, this method calculates the absolute rank of the word. Next, we calculate the average rank score of each word. |
| Log-Rank (Mitchell et al., 2023) | Statistical | Similar to the Rank metric, Log-Rank takes the log probability of the Rank score for each word. |
| Entropy (Mitchell et al., 2023) | Statistical | Similar to the Rank score, Entropy is calculated by obtaining the entropy score of each word, given its context (i.e., previous words), and calculating the average of the final scores. |
| GLTR Test-2 (Rank) (Gehrmann et al., 2019) | Statistical | GLTR uses 3 tests to calculate scores used to distinguish machine-generated text from human-written text. However, following the same procedure used by (He et al., 2023), we only use the 2nd test - calculating the rank of the fraction of words within the top-10, top-100, top-1000, > top-1000 probable words. |
| DetectGPT (Mitchell et al., 2023) | Statistical | DetectGPT perturbs the text and compares the changes between the original and the perturbed text. This comparison is done by calculating the log probability of the original vs. perturbed texts. The hypothesis is that machine-generated text tends to lie in the negative log probability curve, while human-written text will have a higher or lower probability than the perturbed text. |
| RoBERTa-large-OpenAI-detector[14] (Solaiman et al., 2019) | Fine-tuned | This is a sequence classifier based on RoBERTa Large, fine-tuned to distinguish between GPT-2 generated text and WebText. |
| GPT-2 Medium (Radford et al., 2019) | Fine-tuned | GPT-2 Medium[15] is a transformer-based autoregressive language model, pre-trained on English language. |
| XLM-RoBERTa-large (Conneau et al., 2019) | Fine-tuned | XLM-RoBERTa[16] is a pre-trained on 2.5TB of filtered CommonCrawl data containing 100 languages. |
| BERT-base-multilingual-cased (Devlin et al., 2018) | Fine-tuned | Multilingual version of BERT[17] is a masked language model pre-trained on the top 104 languages with the largest Wikipedia. |
| MDeBERTa-v3-base (He et al., 2021a) | Fine-tuned | mDeBERTa[18] is a multilingual version of DeBERTa (He et al., 2021b) trained with CC100 data containing 100 languages. |
| ELECTRA-large (Clark et al., 2020) | Fine-tuned | ELECTRA[19] is a language model pre-trained as a discriminator rather than a generator. |
| mGPT (Shliazhko et al., 2022) | Fine-tuned | mGPT[20] is a multilingual autoregressive model using GPT-3 architecture, trained on 61 languages from 25 language families using Wikipedia and Colossal Clean Crawled Corpus. |

Table 9: A detailed list of all detection methods used in this benchmark together with their descriptions.

# D  Model Parameters

For the purpose of fine-tuning language models for machine-generated text detection task, we have used Trainer[21] API of the transformers library[22] for PyTorch framework. We have used maximum number of 10 epochs with early-stopping mechanism (patience of 5), a batch size of 16 with gradient accumulation of 4 steps, an adaptive learning rate using the AdaFactor optimizer, weight decay of 0.01, half-precision training (except for the mDeBERTa model, where the half-precision training is faulty), using the Macro avg. F1-score as a metric for the best model selection. The tokenizers used truncation of texts to maximum length of 512 tokens. We have done manual hyper-parameter search prior to running the fine-tuning, finding the parameters working for all detector models.

For Random Forest classifier used for entropy-based detector, we have executed optimization of hyperparameters on the train split of the dataset using automated Randomized Grid Search with 5-fold cross-validation and 1k of iterations. The grid consisted of the following parameters:

[14] https://huggingface.co/roberta-large-openai-detector
[15] https://huggingface.co/gpt2-medium
[16] https://huggingface.co/xlm-roberta-large
[17] https://huggingface.co/bert-base-multilingual-cased
[18] https://huggingface.co/microsoft/mdeberta-v3-base
[19] https://huggingface.co/google/electra-large-discriminator
[20] https://huggingface.co/ai-forever/mGPT
[21] https://huggingface.co/docs/transformers/main_classes/trainer
[22] https://github.com/huggingface/transformers

| Detector | ar | ca | cs | de | en | es | nl | pt | ru | uk | zh |
|---|---|---|---|---|---|---|---|---|---|---|---|
| **Entropy + RandomForest** | **0.4860** | 0.4721 | **0.4732** | **0.4729** | 0.4703 | 0.4697 | 0.4692 | 0.4702 | **0.5202** | **0.5040** | 0.4663 |
| **Entropy** | 0.4704 | 0.4705 | 0.4705 | 0.4712 | 0.4706 | 0.4720 | **0.4706** | **0.4716** | 0.4702 | 0.4704 | **0.4704** |
| **Rank** | 0.4704 | 0.4705 | 0.4705 | 0.4712 | 0.4706 | 0.4720 | **0.4706** | **0.4716** | 0.4702 | 0.4704 | **0.4704** |
| **DetectGPT** | 0.4704 | 0.4705 | 0.4705 | 0.4712 | 0.4706 | 0.4720 | **0.4706** | **0.4716** | 0.4702 | 0.4704 | **0.4704** |
| **Log-Rank** | 0.4702 | 0.4705 | 0.4705 | 0.4712 | 0.4706 | 0.4720 | **0.4706** | **0.4716** | 0.4698 | 0.4703 | 0.4644 |
| **Log-likelihood** | 0.4702 | 0.4705 | 0.4705 | 0.4712 | 0.4706 | 0.4720 | **0.4706** | **0.4716** | 0.4699 | 0.4703 | 0.4662 |
| **GLTR Test-2 (Rank)** | 0.4239 | 0.4702 | 0.4700 | 0.4701 | 0.4706 | 0.4720 | 0.4703 | 0.4711 | 0.4697 | 0.4697 | 0.4653 |
| **ZeroGPT** | 0.3055 | **0.4807** | 0.4509 | 0.4019 | **0.5979** | **0.4750** | 0.4625 | 0.4510 | 0.4194 | 0.4267 | 0.1398 |
| **GPTZero** | 0.1128 | 0.1057 | 0.1040 | 0.0999 | 0.5626 | 0.0973 | 0.1044 | 0.1010 | 0.1042 | 0.1014 | 0.1189 |

Table 10: Cross-lingual performance of zero-shot detection models.

- $n\_estimators = [10, 50, 100, 150, 300]$ – a number of trees in the random forest,

- $criterion = ['gini', 'entropy']$ – a function to measure the quality of a split,

- $max\_features = ['sqrt', 'log2', None]$ – a number of features in consideration at every split,

- $max\_depth = [None, 10, 100]$ – a maximum number of levels allowed in each decision tree,

- $min\_samples\_split = [2, 4, 6]$ – a minimum sample number to split a node,

- $min\_samples\_leaf = [1, 3]$ – a minimum sample number that can be stored in a leaf node,

- $bootstrap = [True, False]$ – a method used to sample data points.

## E    Results Data

In Table 10, performance results (Macro average F1-score) of all statistical and black-box detectors per each test language are provided. The highest value for each test language is boldfaced.

In Tables 11–15, performance results (Macro average F1-score) of all fine-tuned detector models per each test language are provided. The highest value in a row is boldfaced. It denotes, on which test language a particular fine-tuned detector version performs the best. At the bottom of the tables, mean values of all detectors per test language are provided, along with separated mean results of multilingual and monolingual detectors' base models.

In Tables 16–18, performance results (Macro average F1-score) of all fine-tuned detector models across individual text-generation LLM data are provided. Also in this case, the highest value in a row is boldfaced.

Table 19 shows how the text-generation LLMs correlate based on the detectors performances, separated per each train language. In Table 20, mean performance results with standard-deviation values are provided for each LLM, aggregated per each train language.

| Train & Test LLM | Detector Base Model | Test Language | | | | | | | | | | |
|---|---|---|---|---|---|---|---|---|---|---|---|---|
| | | ar | ca | cs | de | en | es | nl | pt | ru | uk | zh |
| **alpaca-lora-30b** | bert-base-multilingual-cased | 0.5375 | 0.8271 | 0.8546 | 0.8917 | **0.9567** | 0.7517 | 0.8563 | 0.8055 | 0.8374 | 0.8091 | 0.5537 |
| | electra-large-discriminator | 0.5270 | 0.4929 | 0.4819 | 0.3956 | **0.9783** | 0.5856 | 0.5448 | 0.5196 | 0.5153 | 0.4215 | 0.4642 |
| | gpt2-medium | 0.7438 | 0.4252 | 0.4513 | 0.3928 | **0.9657** | 0.4063 | 0.4233 | 0.3719 | 0.5547 | 0.5326 | 0.3742 |
| | mGPT | 0.4024 | 0.8089 | 0.6132 | 0.8763 | **0.9639** | 0.7390 | 0.8791 | 0.8333 | 0.8162 | 0.8210 | 0.4626 |
| | mdeberta-v3-base | 0.2080 | 0.8592 | 0.7691 | 0.9003 | **0.9439** | 0.7744 | 0.8977 | 0.8570 | 0.7988 | 0.7563 | 0.3099 |
| | roberta-large-openai-detector | 0.3484 | 0.6258 | 0.5430 | 0.5133 | **0.9238** | 0.4809 | 0.7787 | 0.4348 | 0.4029 | 0.4838 | 0.4487 |
| | xlm-roberta-large | 0.4474 | 0.8488 | 0.8713 | 0.9324 | **0.9801** | 0.6319 | 0.7706 | 0.7130 | 0.8733 | 0.8319 | 0.4474 |
| **gpt-3.5-turbo** | bert-base-multilingual-cased | 0.9215 | 0.8904 | 0.9150 | 0.9020 | **0.9783** | 0.8545 | 0.9348 | 0.9124 | 0.9183 | 0.8962 | 0.8933 |
| | electra-large-discriminator | 0.3985 | 0.8409 | 0.3333 | 0.3576 | **0.9765** | 0.8243 | 0.7355 | 0.7810 | 0.3849 | 0.3526 | 0.3399 |
| | gpt2-medium | 0.3673 | 0.3407 | 0.3443 | 0.3369 | **0.9838** | 0.3264 | 0.3432 | 0.3378 | 0.3689 | 0.3710 | 0.3428 |
| | mGPT | 0.7009 | 0.9266 | 0.3370 | 0.9071 | **0.9892** | 0.8506 | 0.9432 | 0.9147 | 0.9013 | 0.9080 | 0.5610 |
| | mdeberta-v3-base | 0.7876 | 0.5599 | 0.7803 | **0.8052** | 0.7178 | 0.6023 | 0.7790 | 0.6782 | 0.6484 | 0.7649 | 0.6733 |
| | roberta-large-openai-detector | 0.4174 | 0.6710 | 0.5623 | 0.5476 | **0.9166** | 0.5356 | 0.8203 | 0.4381 | 0.5442 | 0.6236 | 0.4727 |
| | xlm-roberta-large | 0.7218 | 0.9733 | 0.7211 | 0.9392 | **0.9838** | 0.7861 | 0.9229 | 0.9281 | 0.8839 | 0.6290 | 0.5155 |
| **gpt-4** | bert-base-multilingual-cased | 0.8419 | 0.9265 | 0.7592 | 0.8163 | **0.9765** | 0.8261 | 0.9265 | 0.8704 | 0.7929 | 0.7104 | 0.8360 |
| | electra-large-discriminator | 0.4448 | 0.5611 | 0.3333 | 0.3303 | **0.9928** | 0.7723 | 0.4206 | 0.6814 | 0.4044 | 0.3697 | 0.3333 |
| | gpt2-medium | 0.3853 | 0.3658 | 0.3516 | 0.3593 | **0.9928** | 0.3659 | 0.3887 | 0.3486 | 0.3969 | 0.3732 | 0.3318 |
| | mGPT | 0.6945 | 0.7888 | 0.3866 | 0.8365 | **0.9910** | 0.7954 | 0.8334 | 0.8658 | 0.8008 | 0.7849 | 0.4734 |
| | mdeberta-v3-base | 0.7199 | **0.9733** | 0.8626 | 0.8944 | 0.8255 | 0.9435 | 0.9331 | 0.9319 | 0.8395 | 0.7668 | 0.5706 |
| | roberta-large-openai-detector | 0.5765 | 0.5661 | 0.4808 | 0.5395 | **0.9765** | 0.4974 | 0.7401 | 0.4077 | 0.6384 | 0.5689 | 0.4995 |
| | xlm-roberta-large | 0.5659 | 0.9348 | 0.4987 | 0.8927 | **0.9892** | 0.8592 | 0.9499 | 0.9267 | 0.7676 | 0.4567 | 0.3587 |
| **llama-65b** | bert-base-multilingual-cased | 0.5896 | 0.6070 | 0.6955 | 0.6895 | **0.9017** | 0.7998 | 0.5771 | 0.6752 | 0.7032 | 0.7020 | 0.5194 |
| | electra-large-discriminator | 0.6770 | 0.3311 | 0.3299 | 0.3386 | **0.9220** | 0.3613 | 0.3322 | 0.3538 | 0.4911 | 0.5700 | 0.3281 |
| | gpt2-medium | 0.3735 | 0.4296 | 0.5929 | 0.5370 | **0.8854** | 0.5623 | 0.5372 | 0.3950 | 0.4543 | 0.4893 | 0.4342 |
| | mGPT | 0.3848 | 0.3508 | 0.6898 | 0.7115 | **0.9037** | 0.5248 | 0.4102 | 0.4835 | 0.4669 | 0.4779 | 0.5233 |
| | mdeberta-v3-base | 0.5115 | 0.4060 | 0.8307 | 0.7162 | **0.9003** | 0.6074 | 0.4591 | 0.5413 | 0.7103 | 0.6693 | 0.4384 |
| | roberta-large-openai-detector | 0.3303 | 0.3528 | 0.3446 | 0.3341 | **0.8890** | 0.3460 | 0.3612 | 0.3371 | 0.3311 | 0.3449 | 0.3332 |
| | xlm-roberta-large | 0.4779 | 0.4892 | 0.8507 | 0.7983 | **0.9165** | 0.7241 | 0.4368 | 0.6082 | 0.6899 | 0.6290 | 0.4892 |
| **opt-66b** | bert-base-multilingual-cased | 0.4557 | 0.4586 | 0.6107 | 0.5890 | **0.8592** | 0.5394 | 0.4268 | 0.5635 | 0.5876 | 0.5594 | 0.5041 |
| | electra-large-discriminator | 0.5167 | 0.5157 | 0.6473 | 0.5853 | **0.8899** | 0.6591 | 0.5600 | 0.6993 | 0.4347 | 0.5061 | 0.3273 |
| | gpt2-medium | 0.5226 | 0.4138 | 0.3720 | 0.3827 | **0.8427** | 0.4202 | 0.3647 | 0.4089 | 0.5713 | 0.6213 | 0.4473 |
| | mGPT | 0.2959 | 0.5049 | 0.3954 | 0.6351 | **0.8931** | 0.5064 | 0.4942 | 0.6017 | 0.4416 | 0.4645 | 0.4161 |
| | mdeberta-v3-base | 0.3903 | 0.5181 | 0.6278 | 0.6568 | **0.8538** | 0.6053 | 0.4927 | 0.6431 | 0.5799 | 0.6016 | 0.5617 |
| | roberta-large-openai-detector | 0.3315 | 0.5268 | 0.4635 | 0.4038 | **0.8711** | 0.3792 | 0.5761 | 0.3714 | 0.3293 | 0.3733 | 0.3591 |
| | xlm-roberta-large | 0.4910 | 0.4613 | 0.5495 | 0.6735 | **0.8827** | 0.5613 | 0.4187 | 0.5897 | 0.6706 | 0.6961 | 0.5358 |
| **opt-iml-max-1.3b** | bert-base-multilingual-cased | 0.4592 | 0.5171 | 0.5792 | 0.5916 | **0.9061** | 0.5143 | 0.5136 | 0.4721 | 0.5494 | 0.5333 | 0.4390 |
| | electra-large-discriminator | 0.3264 | 0.5542 | 0.5036 | 0.6618 | **0.9819** | 0.8709 | 0.7625 | 0.8334 | 0.3309 | 0.3390 | 0.3299 |
| | gpt2-medium | 0.3869 | 0.7116 | 0.5300 | 0.5259 | **0.9368** | 0.5789 | 0.5600 | 0.5453 | 0.4733 | 0.5295 | 0.4720 |
| | mGPT | 0.3380 | 0.5134 | 0.6410 | 0.7450 | **0.9585** | 0.5284 | 0.6955 | 0.5212 | 0.4829 | 0.4925 | 0.5788 |
| | mdeberta-v3-base | 0.7932 | 0.9731 | **1.0000** | 0.9609 | 0.9257 | 0.9023 | 0.9615 | 0.9025 | 0.9471 | 0.9570 | 0.7016 |
| | roberta-large-openai-detector | 0.3427 | 0.8511 | 0.7842 | 0.6490 | **0.9729** | 0.6895 | 0.9211 | 0.6590 | 0.3614 | 0.4318 | 0.4777 |
| | xlm-roberta-large | 0.5810 | 0.8567 | 0.8657 | 0.8416 | **0.9439** | 0.7268 | 0.7684 | 0.6991 | 0.7483 | 0.7587 | 0.6237 |
| **text-davinci-003** | bert-base-multilingual-cased | 0.7647 | 0.9014 | 0.8537 | 0.8554 | **0.9693** | 0.7750 | 0.9032 | 0.8408 | 0.7161 | 0.6657 | 0.8409 |
| | electra-large-discriminator | 0.4628 | 0.5751 | 0.4275 | 0.4024 | **0.9783** | 0.7471 | 0.5568 | 0.6000 | 0.5075 | 0.4826 | 0.4880 |
| | gpt2-medium | 0.3322 | 0.3333 | 0.3333 | 0.3303 | **0.9856** | 0.3264 | 0.3367 | 0.3268 | 0.3267 | 0.3361 | 0.3326 |
| | mGPT | 0.5393 | 0.9199 | 0.3623 | 0.8936 | **0.9783** | 0.7678 | 0.9027 | 0.8389 | 0.7713 | 0.6932 | 0.5246 |
| | mdeberta-v3-base | 0.5432 | 0.6398 | 0.8239 | 0.8563 | **0.9513** | 0.5774 | 0.7596 | 0.6335 | 0.6969 | 0.6015 | 0.7779 |
| | roberta-large-openai-detector | 0.4087 | 0.5610 | 0.5307 | 0.5112 | **0.9293** | 0.4369 | 0.7561 | 0.3902 | 0.4746 | 0.6221 | 0.4820 |
| | xlm-roberta-large | 0.5035 | 0.9213 | 0.6965 | 0.8913 | **0.9856** | 0.7715 | 0.8973 | 0.8232 | 0.5769 | 0.4739 | 0.4377 |
| **vicuna-13b** | bert-base-multilingual-cased | 0.7913 | 0.8856 | 0.8012 | 0.8446 | **0.9747** | 0.7859 | 0.9365 | 0.8520 | 0.8309 | 0.8137 | 0.6667 |
| | electra-large-discriminator | 0.3912 | 0.3848 | 0.6025 | 0.5077 | **0.9838** | 0.6577 | 0.4839 | 0.5162 | 0.3420 | 0.3296 | 0.3551 |
| | gpt2-medium | 0.6275 | 0.3658 | 0.4420 | 0.4286 | **0.9783** | 0.3931 | 0.3921 | 0.3814 | 0.4329 | 0.4687 | 0.3502 |
| | mGPT | 0.4086 | 0.7888 | 0.5043 | 0.8511 | **0.9765** | 0.6633 | 0.8876 | 0.8071 | 0.6347 | 0.7088 | 0.5941 |
| | mdeberta-v3-base | 0.1784 | 0.5000 | 0.5018 | 0.6295 | **0.7775** | 0.4754 | 0.6191 | 0.5078 | 0.4121 | 0.4635 | 0.5888 |
| | roberta-large-openai-detector | 0.4179 | 0.6409 | 0.5661 | 0.5341 | **0.9348** | 0.5070 | 0.8146 | 0.4350 | 0.5180 | 0.5303 | 0.4740 |
| | xlm-roberta-large | 0.3864 | 0.9417 | 0.4894 | 0.9086 | **0.9801** | 0.6866 | 0.8400 | 0.7163 | 0.7787 | 0.4897 | 0.4370 |
| | **All Detectors Mean** | 0.5016 | 0.6412 | 0.5909 | 0.6578 | 0.9361 | 0.6284 | 0.6703 | 0.6273 | 0.5976 | 0.5832 | 0.4902 |
| | **Multilingual Base Models Mean** | 0.5448 | 0.7335 | 0.6793 | 0.8104 | 0.9292 | 0.7018 | 0.7508 | 0.7362 | 0.7148 | 0.6746 | 0.5580 |
| | **Monolingual Base Models Mean** | 0.4440 | 0.5182 | 0.4730 | 0.4544 | 0.9454 | 0.5304 | 0.5629 | 0.4822 | 0.4412 | 0.4613 | 0.3999 |

Table 11: Cross-lingual performance of all detection models fine-tuned on the English language (evaluated per LLM).

| Train & Test LLM | Detector Base Model | Test Language | | | | | | | | | | |
|---|---|---|---|---|---|---|---|---|---|---|---|---|
| | | ar | ca | cs | de | en | es | nl | pt | ru | uk | zh |
| **alpaca-lora-30b** | bert-base-multilingual-cased | 0.6019 | 0.8950 | 0.8108 | 0.8765 | 0.7517 | **0.9092** | 0.8941 | 0.8943 | 0.8600 | 0.8182 | 0.7713 |
| | electra-large-discriminator | 0.5116 | 0.4754 | 0.3370 | 0.3361 | 0.3325 | **0.9348** | 0.5286 | 0.8665 | 0.3891 | 0.3820 | 0.3986 |
| | gpt2-medium | 0.6937 | 0.8090 | 0.6564 | 0.8226 | 0.7332 | **0.8935** | 0.8291 | 0.8602 | 0.5735 | 0.5679 | 0.4391 |
| | mGPT | 0.6980 | 0.8639 | 0.6989 | 0.9001 | 0.6589 | **0.9452** | 0.7088 | 0.9403 | 0.8366 | 0.8880 | 0.5691 |
| | mdeberta-v3-base | 0.8255 | 0.7338 | 0.9110 | 0.9171 | 0.8771 | 0.8256 | 0.8038 | 0.7777 | **0.9317** | 0.9096 | 0.7954 |
| | roberta-large-openai-detector | 0.3578 | 0.7052 | 0.6789 | 0.8414 | 0.2717 | **0.9295** | 0.8296 | 0.8671 | 0.3587 | 0.3634 | 0.7537 |
| | xlm-roberta-large | 0.6999 | 0.8349 | 0.9008 | 0.9493 | 0.9295 | 0.9281 | **0.9532** | 0.9301 | 0.8845 | 0.8278 | 0.6164 |
| **gpt-3.5-turbo** | bert-base-multilingual-cased | 0.8896 | **0.9365** | 0.8997 | 0.8900 | 0.9169 | 0.9263 | 0.9232 | 0.8858 | 0.8949 | 0.8722 | 0.8765 |
| | electra-large-discriminator | 0.5268 | 0.7374 | 0.3333 | 0.3340 | 0.3333 | **0.9743** | 0.4290 | 0.9215 | 0.5390 | 0.4383 | 0.3333 |
| | gpt2-medium | 0.4313 | 0.8889 | 0.3900 | 0.3827 | 0.3669 | **0.8970** | 0.4874 | 0.8320 | 0.4949 | 0.4526 | 0.3355 |
| | mGPT | 0.8941 | 0.9650 | 0.3552 | 0.9307 | 0.8397 | **0.9692** | 0.8974 | 0.9556 | 0.8942 | 0.9196 | 0.6781 |
| | mdeberta-v3-base | 0.9281 | 0.8817 | **0.9666** | 0.9577 | 0.9368 | 0.9570 | 0.9109 | 0.9089 | 0.8890 | 0.9128 | 0.9416 |
| | roberta-large-openai-detector | 0.3307 | 0.8610 | 0.8449 | 0.8700 | 0.2073 | **0.9153** | 0.8743 | 0.8392 | 0.3333 | 0.3311 | 0.7601 |
| | xlm-roberta-large | 0.7671 | 0.9550 | 0.8609 | 0.8539 | **0.9801** | 0.9743 | 0.9616 | 0.9676 | 0.9093 | 0.7368 | 0.8504 |
| **gpt-4** | bert-base-multilingual-cased | 0.8764 | 0.8852 | **0.9300** | 0.8808 | 0.7306 | 0.9277 | 0.9075 | 0.8679 | 0.8916 | 0.8792 | 0.8799 |
| | electra-large-discriminator | 0.3860 | 0.8103 | 0.3333 | 0.3303 | 0.3373 | **0.9571** | 0.5424 | 0.9506 | 0.3798 | 0.3670 | 0.3407 |
| | gpt2-medium | 0.4260 | 0.7906 | 0.4326 | 0.4318 | 0.3362 | **0.9143** | 0.4571 | 0.7844 | 0.4378 | 0.4042 | 0.3399 |
| | mGPT | 0.8491 | 0.8705 | 0.3370 | 0.9187 | 0.9476 | 0.9829 | 0.9245 | **0.9830** | 0.9016 | 0.8454 | 0.6649 |
| | mdeberta-v3-base | 0.9331 | 0.8897 | **0.9549** | 0.9356 | 0.8826 | 0.9195 | 0.9331 | 0.9027 | 0.8545 | 0.9096 | 0.9045 |
| | roberta-large-openai-detector | 0.3322 | 0.9128 | 0.8091 | 0.8765 | 0.3602 | **0.9293** | 0.8449 | 0.8768 | 0.3370 | 0.3318 | 0.6283 |
| | xlm-roberta-large | 0.5124 | 0.9599 | 0.5688 | 0.8011 | 0.9729 | **0.9812** | 0.9616 | 0.9676 | 0.5987 | 0.3994 | 0.4001 |
| **llama-65b** | bert-base-multilingual-cased | 0.7956 | 0.9177 | 0.9645 | 0.7711 | 0.3611 | **0.9914** | 0.6665 | 0.9794 | 0.8227 | 0.9290 | 0.6396 |
| | electra-large-discriminator | 0.4887 | 0.9026 | 0.9272 | 0.9214 | 0.3325 | **0.9741** | 0.8515 | 0.9691 | 0.3620 | 0.4606 | 0.3318 |
| | gpt2-medium | 0.3680 | 0.9365 | 0.8738 | 0.9079 | 0.3990 | **0.9621** | 0.8744 | 0.9398 | 0.4793 | 0.5608 | 0.5344 |
| | mGPT | 0.5912 | 0.6899 | 0.8606 | 0.7540 | 0.3505 | **0.9673** | 0.3604 | 0.9501 | 0.5401 | 0.8348 | 0.3731 |
| | mdeberta-v3-base | 0.9135 | 0.9262 | **0.9882** | 0.9608 | 0.6260 | 0.9862 | 0.6591 | 0.9777 | 0.9765 | 0.9848 | 0.3539 |
| | roberta-large-openai-detector | 0.3296 | 0.8944 | 0.7558 | 0.7449 | 0.3357 | **0.9517** | 0.8188 | 0.8808 | 0.3392 | 0.3617 | 0.3296 |
| | xlm-roberta-large | 0.8461 | 0.8307 | 0.9831 | 0.9557 | 0.6680 | **0.9879** | 0.4583 | 0.9725 | 0.9412 | 0.9680 | 0.6842 |
| **opt-66b** | bert-base-multilingual-cased | 0.7378 | 0.7608 | 0.6443 | 0.5872 | 0.4705 | 0.7658 | 0.7038 | 0.7662 | 0.6748 | 0.7543 | **0.8225** |
| | electra-large-discriminator | 0.4594 | 0.7202 | 0.6132 | 0.6954 | 0.3333 | **0.7692** | 0.6844 | 0.7265 | 0.4311 | 0.5203 | 0.3311 |
| | gpt2-medium | 0.4455 | 0.7310 | 0.4703 | 0.5913 | 0.4524 | **0.7703** | 0.5466 | 0.5972 | 0.5993 | 0.6464 | 0.4690 |
| | mGPT | 0.3906 | 0.7297 | 0.4550 | 0.6957 | 0.3722 | **0.7992** | 0.3866 | 0.5949 | 0.5713 | 0.6426 | 0.4554 |
| | mdeberta-v3-base | 0.9139 | 0.8500 | **0.9883** | 0.9493 | 0.6240 | 0.8762 | 0.8538 | 0.7872 | 0.9696 | 0.9627 | 0.6997 |
| | roberta-large-openai-detector | 0.4668 | 0.6129 | 0.8550 | 0.6502 | 0.3355 | 0.8141 | **0.8564** | 0.7337 | 0.5548 | 0.5646 | 0.5596 |
| | xlm-roberta-large | 0.6058 | 0.7452 | **0.8765** | 0.8104 | 0.4443 | 0.8123 | 0.4702 | 0.6335 | 0.8106 | 0.8408 | 0.5381 |
| **opt-iml-max-1.3b** | bert-base-multilingual-cased | 0.9537 | 0.9697 | 0.9632 | 0.8676 | 0.6840 | 0.9482 | 0.9699 | 0.8820 | 0.9027 | **0.9707** | 0.9528 |
| | electra-large-discriminator | 0.3527 | **0.9681** | 0.8392 | 0.8930 | 0.3333 | 0.9515 | 0.8798 | 0.9189 | 0.3280 | 0.3305 | 0.3307 |
| | gpt2-medium | 0.6084 | 0.9049 | 0.7598 | 0.5085 | 0.4730 | **0.9100** | 0.6565 | 0.6696 | 0.6559 | 0.7088 | 0.3941 |
| | mGPT | 0.9434 | 0.9613 | 0.9210 | **0.9932** | 0.4045 | 0.9879 | 0.8410 | 0.9810 | 0.8970 | 0.9845 | 0.5838 |
| | mdeberta-v3-base | 0.9931 | 0.9866 | 0.9933 | 0.8710 | 0.5865 | 0.9827 | 0.9329 | 0.9828 | **0.9949** | 0.9931 | 0.8744 |
| | roberta-large-openai-detector | 0.4128 | **0.9646** | 0.9498 | 0.7265 | 0.2160 | 0.9602 | 0.9381 | 0.9103 | 0.5456 | 0.6839 | 0.5212 |
| | xlm-roberta-large | 0.9794 | **0.9916** | 0.8583 | 0.9881 | 0.6192 | 0.9810 | 0.9799 | 0.9793 | 0.7578 | 0.7060 | 0.8187 |
| **text-davinci-003** | bert-base-multilingual-cased | 0.7319 | 0.9047 | 0.8833 | 0.8602 | 0.7605 | **0.9312** | 0.8892 | 0.8631 | 0.8027 | 0.7405 | 0.8400 |
| | electra-large-discriminator | 0.4855 | 0.5814 | 0.3333 | 0.3296 | 0.3373 | **0.9605** | 0.4529 | 0.8925 | 0.5528 | 0.5678 | 0.3370 |
| | gpt2-medium | 0.6145 | **0.9250** | 0.6295 | 0.7554 | 0.5875 | 0.9047 | 0.7750 | 0.8313 | 0.7541 | 0.6885 | 0.3355 |
| | mGPT | 0.7188 | 0.8020 | 0.3832 | 0.9274 | 0.9039 | **0.9657** | 0.9382 | 0.9165 | 0.8578 | 0.7165 | 0.7663 |
| | mdeberta-v3-base | 0.6436 | 0.7615 | 0.9248 | 0.9239 | **0.9404** | 0.8059 | 0.9065 | 0.8484 | 0.8597 | 0.7446 | 0.9166 |
| | roberta-large-openai-detector | 0.3329 | 0.5973 | 0.7689 | 0.8681 | 0.2113 | **0.9415** | 0.7787 | 0.9003 | 0.3337 | 0.3388 | 0.7381 |
| | xlm-roberta-large | 0.7993 | 0.9264 | 0.9482 | 0.9510 | 0.9422 | **0.9777** | 0.9733 | 0.9352 | 0.8944 | 0.7921 | 0.8768 |
| **vicuna-13b** | bert-base-multilingual-cased | 0.8628 | 0.9149 | 0.7705 | 0.8341 | 0.7503 | **0.9281** | 0.8996 | 0.8823 | 0.8226 | 0.7869 | 0.8514 |
| | electra-large-discriminator | 0.4812 | 0.5839 | 0.3798 | 0.3827 | 0.3333 | **0.9743** | 0.4827 | 0.8677 | 0.5833 | 0.5508 | 0.4081 |
| | gpt2-medium | 0.7679 | **0.9232** | 0.5873 | 0.6943 | 0.7214 | 0.9159 | 0.7333 | 0.8299 | 0.5051 | 0.5253 | 0.4208 |
| | mGPT | 0.7234 | 0.8716 | 0.5427 | 0.9206 | 0.6739 | **0.9674** | 0.6880 | 0.9164 | 0.7205 | 0.7893 | 0.6191 |
| | mdeberta-v3-base | 0.6784 | 0.8994 | 0.8387 | 0.9052 | 0.9169 | **0.9520** | 0.8753 | 0.8956 | 0.8595 | 0.8562 | 0.8291 |
| | roberta-large-openai-detector | 0.5358 | 0.8853 | 0.6725 | 0.8681 | 0.3147 | **0.9432** | 0.8460 | 0.9058 | 0.4371 | 0.3755 | 0.5963 |
| | xlm-roberta-large | 0.8447 | 0.8789 | 0.6681 | 0.8614 | 0.9061 | **0.9434** | 0.8267 | 0.8959 | 0.7773 | 0.6400 | 0.6473 |
| | **All Detectors Mean** | 0.6480 | 0.8413 | 0.7300 | 0.7850 | 0.5790 | 0.9259 | 0.7689 | 0.8749 | 0.6804 | 0.6800 | 0.6082 |
| | **Multilingual Base Models Mean** | 0.7857 | 0.8747 | 0.8016 | 0.8812 | 0.7322 | 0.9314 | 0.8143 | 0.8944 | 0.8375 | 0.8299 | 0.7216 |
| | **Monolingual Base Models Mean** | 0.4644 | 0.7967 | 0.6346 | 0.6568 | 0.3748 | 0.9187 | 0.7082 | 0.8488 | 0.4710 | 0.4801 | 0.4569 |

Table 12: Cross-lingual performance of all detection models fine-tuned on the Spanish language (evaluated per LLM).

| Train & Test LLM | Detector Base Model | Test Language | | | | | | | | | | |
|---|---|---|---|---|---|---|---|---|---|---|---|---|
| | | ar | ca | cs | de | en | es | nl | pt | ru | uk | zh |
| alpaca-lora-30b | bert-base-multilingual-cased | 0.6026 | 0.6327 | 0.8067 | 0.7409 | 0.4634 | 0.6572 | 0.6705 | 0.6567 | **0.9099** | 0.8829 | 0.6696 |
| | electra-large-discriminator | 0.4945 | 0.3680 | 0.3443 | 0.3516 | 0.3829 | 0.5214 | 0.4081 | 0.3884 | **0.5506** | 0.5100 | 0.4514 |
| | gpt2-medium | 0.7291 | 0.4784 | 0.4665 | 0.4094 | 0.6639 | 0.6505 | 0.4265 | 0.5331 | **0.8599** | 0.8249 | 0.3950 |
| | mGPT | 0.5635 | 0.4123 | 0.6725 | 0.8716 | 0.5272 | 0.7714 | 0.5325 | 0.7920 | **0.9350** | 0.9129 | 0.4906 |
| | mdeberta-v3-base | 0.9716 | 0.8123 | **0.9850** | 0.9409 | 0.8550 | 0.8885 | 0.8868 | 0.8199 | 0.9617 | 0.9615 | 0.8237 |
| | roberta-large-openai-detector | 0.7062 | 0.3326 | 0.3407 | 0.3363 | 0.2381 | 0.3394 | 0.3370 | 0.3421 | **0.8599** | 0.8464 | 0.5544 |
| | xlm-roberta-large | 0.9180 | 0.8983 | **0.9700** | 0.9442 | 0.9169 | 0.9058 | 0.9431 | 0.8993 | 0.9433 | 0.9397 | 0.8522 |
| gpt-3.5-turbo | bert-base-multilingual-cased | 0.9416 | 0.8680 | 0.9045 | 0.8538 | 0.7810 | 0.8754 | 0.8783 | 0.8378 | **0.9567** | 0.9331 | 0.8666 |
| | electra-large-discriminator | **0.6636** | 0.3333 | 0.3333 | 0.3303 | 0.3457 | 0.3272 | 0.3330 | 0.3276 | 0.6336 | 0.5592 | 0.3333 |
| | gpt2-medium | 0.5384 | 0.3443 | 0.3407 | 0.3369 | 0.3423 | 0.3264 | 0.3403 | 0.3350 | **0.8716** | 0.8061 | 0.3333 |
| | mGPT | 0.7846 | 0.8857 | 0.3552 | 0.9070 | 0.8817 | 0.9228 | 0.9230 | 0.9043 | **0.9533** | 0.9244 | 0.6258 |
| | mdeberta-v3-base | 0.9365 | 0.8387 | **0.9817** | 0.9256 | 0.9603 | 0.9296 | 0.9146 | 0.9023 | 0.9283 | 0.9482 | 0.8329 |
| | roberta-large-openai-detector | 0.3496 | 0.3472 | 0.3720 | 0.3476 | 0.1697 | 0.3471 | 0.3663 | 0.3421 | **0.8695** | 0.5855 | 0.6240 |
| | xlm-roberta-large | 0.9332 | 0.9432 | **0.9883** | 0.9391 | 0.9675 | 0.9435 | 0.9466 | 0.9143 | 0.9767 | 0.9615 | 0.9048 |
| gpt-4 | bert-base-multilingual-cased | 0.8667 | 0.7067 | 0.7828 | 0.7535 | 0.4958 | 0.6805 | 0.7514 | 0.6918 | **0.9333** | 0.9214 | 0.8122 |
| | electra-large-discriminator | 0.5003 | 0.3311 | 0.3333 | 0.3303 | 0.3718 | 0.3374 | 0.3292 | 0.3326 | **0.6837** | 0.6241 | 0.3677 |
| | gpt2-medium | 0.5234 | 0.3587 | 0.3623 | 0.3514 | 0.3368 | 0.3659 | 0.3655 | 0.3458 | 0.7535 | **0.7794** | 0.3508 |
| | mGPT | 0.8978 | 0.8978 | 0.3587 | 0.9188 | 0.8713 | 0.9383 | 0.9246 | 0.9264 | **0.9566** | 0.8570 | 0.7078 |
| | mdeberta-v3-base | **0.9349** | 0.5556 | 0.8788 | 0.9221 | 0.9259 | 0.7611 | 0.7609 | 0.6932 | 0.8305 | 0.8940 | 0.7797 |
| | roberta-large-openai-detector | 0.5936 | 0.3407 | 0.3480 | 0.3439 | 0.2994 | 0.3433 | 0.3628 | 0.3421 | 0.7991 | **0.8526** | 0.6649 |
| | xlm-roberta-large | 0.9215 | 0.9600 | **0.9950** | 0.9391 | 0.9765 | 0.9349 | 0.9633 | 0.9368 | 0.9567 | 0.9447 | 0.9214 |
| llama-65b | bert-base-multilingual-cased | 0.7078 | 0.5300 | 0.8750 | 0.6477 | 0.3325 | 0.8211 | 0.5191 | 0.6043 | **0.9849** | 0.9764 | 0.4765 |
| | electra-large-discriminator | **0.7140** | 0.3303 | 0.3299 | 0.3348 | 0.3325 | 0.3383 | 0.3330 | 0.3356 | 0.6109 | 0.6025 | 0.3676 |
| | gpt2-medium | 0.6015 | 0.4216 | 0.6322 | 0.6629 | 0.4170 | 0.7424 | 0.5012 | 0.4982 | **0.9395** | 0.9394 | 0.5690 |
| | mGPT | 0.5347 | 0.3326 | 0.6154 | 0.6470 | 0.3325 | 0.4964 | 0.3404 | 0.4847 | **0.9782** | 0.9359 | 0.4056 |
| | mdeberta-v3-base | 0.7629 | 0.4277 | 0.9848 | 0.8540 | 0.3824 | 0.7305 | 0.4145 | 0.5383 | **0.9883** | 0.9815 | 0.5622 |
| | roberta-large-openai-detector | 0.6454 | 0.3311 | 0.3269 | 0.3341 | 0.3388 | 0.3375 | 0.3307 | 0.3356 | **0.9631** | 0.9309 | 0.3969 |
| | xlm-roberta-large | 0.8858 | 0.5705 | 0.9865 | 0.9160 | 0.3705 | 0.5024 | 0.3674 | 0.5826 | **0.9933** | 0.9882 | 0.6314 |
| opt-66b | bert-base-multilingual-cased | 0.7593 | 0.6061 | 0.7673 | 0.6078 | 0.4720 | 0.6423 | 0.6428 | 0.6734 | **0.9340** | 0.9254 | 0.6408 |
| | electra-large-discriminator | 0.5543 | 0.3326 | 0.3333 | 0.3356 | 0.3325 | 0.3394 | 0.3337 | 0.3375 | **0.7304** | 0.6932 | 0.3425 |
| | gpt2-medium | 0.6817 | 0.3755 | 0.3814 | 0.3576 | 0.4726 | 0.3972 | 0.3388 | 0.3598 | **0.9085** | 0.8797 | 0.4361 |
| | mGPT | 0.9714 | 0.5123 | 0.5776 | 0.3999 | 0.3473 | 0.4262 | 0.6050 | 0.4165 | 0.9662 | **0.9814** | 0.7649 |
| | mdeberta-v3-base | 0.9393 | 0.7428 | 0.9833 | 0.4860 | 0.3413 | 0.3394 | 0.6255 | 0.3610 | 0.9848 | **0.9932** | 0.7645 |
| | roberta-large-openai-detector | 0.4195 | 0.3266 | 0.3326 | 0.3348 | 0.3046 | 0.3386 | 0.3322 | 0.3375 | **0.9003** | 0.8830 | 0.3759 |
| | xlm-roberta-large | 0.9546 | 0.8552 | **0.9917** | 0.5544 | 0.3453 | 0.5474 | 0.7734 | 0.6168 | 0.9848 | 0.9864 | 0.8530 |
| opt-iml-max-1.3b | bert-base-multilingual-cased | 0.9588 | 0.5420 | 0.8707 | 0.5073 | 0.3333 | 0.4417 | 0.5379 | 0.4153 | 0.9795 | **0.9828** | 0.8243 |
| | electra-large-discriminator | 0.6770 | 0.3709 | 0.5542 | 0.4481 | 0.3669 | 0.5480 | 0.4653 | 0.5424 | 0.8379 | **0.8536** | 0.7043 |
| | gpt2-medium | 0.8338 | 0.5143 | 0.4434 | 0.3979 | 0.3991 | 0.4127 | 0.4003 | 0.3891 | **0.9368** | 0.9188 | 0.5273 |
| | mGPT | 0.9640 | 0.6583 | 0.3597 | 0.5613 | 0.3750 | 0.3873 | 0.8374 | 0.4378 | **0.9949** | 0.9810 | 0.7678 |
| | mdeberta-v3-base | 0.8902 | 0.5114 | 0.9379 | 0.5767 | 0.3333 | 0.3926 | 0.6230 | 0.3984 | 0.9224 | **0.9707** | 0.8335 |
| | roberta-large-openai-detector | 0.7226 | 0.3149 | 0.3979 | 0.3242 | 0.1294 | 0.3216 | 0.3521 | 0.3245 | **0.9333** | 0.9293 | 0.5404 |
| | xlm-roberta-large | 0.9931 | 0.7275 | 0.9900 | 0.7551 | 0.3373 | 0.3944 | 0.8638 | 0.4432 | **0.9966** | 0.9948 | 0.9338 |
| text-davinci-003 | bert-base-multilingual-cased | 0.7480 | 0.6998 | 0.6476 | 0.8087 | 0.7249 | 0.8167 | 0.7157 | 0.7645 | **0.9146** | 0.9079 | 0.7911 |
| | electra-large-discriminator | 0.4395 | 0.3348 | 0.3326 | 0.3332 | **0.5357** | 0.3413 | 0.3306 | 0.3318 | 0.5136 | 0.4716 | 0.3935 |
| | gpt2-medium | 0.7234 | 0.3333 | 0.3333 | 0.3296 | 0.3244 | 0.3264 | 0.3367 | 0.3245 | **0.8528** | 0.8408 | 0.3333 |
| | mGPT | 0.8798 | 0.6376 | 0.3552 | 0.8629 | 0.6831 | 0.8696 | 0.7987 | 0.7996 | **0.9315** | 0.8193 | 0.6410 |
| | mdeberta-v3-base | 0.8800 | 0.4399 | 0.6877 | 0.5837 | 0.3453 | 0.5184 | 0.4845 | 0.5132 | 0.9381 | **0.9598** | 0.7358 |
| | roberta-large-openai-detector | 0.6924 | 0.3623 | 0.4165 | 0.3514 | 0.1657 | 0.3471 | 0.4236 | 0.3421 | **0.8814** | 0.8155 | 0.6138 |
| | xlm-roberta-large | 0.8664 | 0.4440 | 0.7422 | 0.7117 | 0.3778 | 0.5679 | 0.5897 | 0.5700 | 0.9382 | **0.9497** | 0.7750 |
| vicuna-13b | bert-base-multilingual-cased | 0.5450 | 0.3623 | 0.5724 | 0.4549 | 0.3333 | 0.4751 | 0.3548 | 0.3721 | **0.9264** | 0.8401 | 0.3472 |
| | electra-large-discriminator | 0.5084 | 0.3355 | 0.3376 | 0.3318 | 0.3333 | 0.3386 | 0.3306 | 0.3421 | **0.5535** | 0.5256 | 0.5498 |
| | gpt2-medium | 0.5858 | 0.3787 | 0.4784 | 0.4744 | 0.5022 | 0.3811 | 0.3954 | 0.4326 | **0.8712** | 0.6147 | 0.4456 |
| | mGPT | 0.6846 | 0.3407 | 0.8659 | 0.8576 | 0.3722 | 0.6188 | 0.3663 | 0.5499 | **0.9548** | 0.8978 | 0.5443 |
| | mdeberta-v3-base | **0.9883** | 0.7980 | 0.9381 | 0.9033 | 0.8136 | 0.9023 | 0.8760 | 0.8464 | 0.9565 | 0.9582 | 0.8594 |
| | roberta-large-openai-detector | 0.5037 | 0.3326 | 0.3318 | 0.3356 | 0.3138 | 0.3394 | 0.3330 | 0.3382 | **0.8845** | 0.5946 | 0.6119 |
| | xlm-roberta-large | **0.9716** | 0.7526 | 0.9280 | 0.9391 | 0.8572 | 0.9298 | 0.6979 | 0.8416 | 0.9615 | 0.9278 | 0.8998 |
| | **All Detectors Mean** | 0.7421 | 0.5274 | 0.6171 | 0.5914 | 0.4795 | 0.5614 | 0.5524 | 0.5369 | 0.8870 | 0.8557 | 0.6183 |
| | **Multilingual Base Models Mean** | 0.8487 | 0.6532 | 0.7924 | 0.7591 | 0.5760 | 0.6884 | 0.6915 | 0.6626 | 0.9522 | 0.9387 | 0.7294 |
| | **Monolingual Base Models Mean** | 0.6001 | 0.3596 | 0.3835 | 0.3677 | 0.3508 | 0.3920 | 0.3669 | 0.3692 | 0.8000 | 0.7451 | 0.4701 |

Table 13: Cross-lingual performance of all detection models fine-tuned on the Russian language (evaluated per LLM).

| Train & Test LLM | Detector Base Model | Test Language | | | | | | | | | | |
|---|---|---|---|---|---|---|---|---|---|---|---|---|
| | | ar | ca | cs | de | en | es | nl | pt | ru | uk | zh |
| **alpaca-lora-30b** | bert-base-multilingual-cased | 0.5096 | 0.9133 | 0.8879 | 0.9138 | **0.9657** | 0.9401 | 0.9161 | 0.9113 | 0.9200 | 0.8928 | 0.7472 |
| | electra-large-discriminator | 0.3367 | 0.7431 | 0.4389 | 0.5051 | **0.9856** | 0.9502 | 0.7716 | 0.8824 | 0.3407 | 0.3326 | 0.3999 |
| | gpt2-medium | 0.8118 | 0.7896 | 0.6623 | 0.7747 | **0.9819** | 0.8919 | 0.8064 | 0.8307 | 0.8463 | 0.8251 | 0.4282 |
| | mGPT | 0.6955 | 0.8833 | 0.6480 | 0.9290 | **0.9585** | 0.9315 | 0.8215 | 0.9334 | 0.9383 | 0.9129 | 0.5555 |
| | mdeberta-v3-base | 0.8438 | 0.8062 | 0.8906 | 0.8727 | 0.9075 | 0.8643 | 0.8044 | 0.8364 | 0.9283 | **0.9331** | 0.7118 |
| | roberta-large-openai-detector | 0.8040 | 0.7010 | 0.8484 | 0.8629 | 0.9348 | **0.9363** | 0.8641 | 0.8969 | 0.8817 | 0.8487 | 0.7252 |
| | xlm-roberta-large | 0.7742 | 0.9283 | 0.9600 | 0.9578 | **0.9856** | 0.9589 | 0.9481 | 0.9522 | 0.9366 | 0.9246 | 0.8200 |
| **gpt-3.5-turbo** | bert-base-multilingual-cased | 0.8483 | 0.9750 | 0.9633 | 0.9391 | **0.9819** | 0.9777 | 0.9816 | 0.9506 | 0.9550 | 0.9313 | 0.9148 |
| | electra-large-discriminator | 0.3501 | 0.7954 | 0.3333 | 0.3340 | **0.9819** | 0.9657 | 0.4354 | 0.8583 | 0.3326 | 0.3326 | 0.3901 |
| | gpt2-medium | 0.6911 | 0.9180 | 0.5441 | 0.6283 | **0.9874** | 0.9452 | 0.7647 | 0.8586 | 0.8480 | 0.6995 | 0.3333 |
| | mGPT | 0.8856 | 0.9198 | 0.3552 | 0.8964 | **0.9801** | 0.9469 | 0.9566 | 0.9370 | 0.9383 | 0.8939 | 0.5693 |
| | mdeberta-v3-base | 0.9466 | 0.8710 | **0.9767** | 0.9578 | 0.9747 | 0.9280 | 0.8977 | 0.9039 | 0.9232 | 0.9398 | 0.8526 |
| | roberta-large-openai-detector | 0.5451 | 0.6366 | 0.6189 | 0.7365 | **0.9765** | 0.9726 | 0.5633 | 0.9710 | 0.8933 | 0.5315 | 0.4295 |
| | xlm-roberta-large | 0.9280 | 0.9850 | 0.9883 | 0.9425 | **0.9910** | 0.9846 | 0.9800 | 0.9727 | 0.9850 | 0.9464 | 0.9382 |
| **gpt-4** | bert-base-multilingual-cased | 0.9416 | 0.9566 | 0.9533 | 0.9255 | **0.9928** | 0.9485 | 0.9683 | 0.9453 | 0.9400 | 0.9365 | 0.9048 |
| | electra-large-discriminator | 0.3330 | 0.9297 | 0.3333 | 0.3593 | **0.9946** | 0.9726 | 0.8213 | 0.9472 | 0.3333 | 0.3326 | 0.3333 |
| | gpt2-medium | 0.7644 | 0.8923 | 0.6236 | 0.6851 | **0.9982** | 0.9279 | 0.7007 | 0.8714 | 0.8516 | 0.8235 | 0.3392 |
| | mGPT | 0.8911 | 0.9450 | 0.3407 | 0.9014 | **0.9928** | 0.9640 | 0.9246 | 0.9487 | 0.9467 | 0.9212 | 0.7236 |
| | mdeberta-v3-base | 0.9382 | 0.9315 | **0.9883** | 0.9797 | 0.9457 | 0.9502 | 0.9582 | 0.9195 | 0.9383 | 0.9498 | 0.8854 |
| | roberta-large-openai-detector | 0.7820 | 0.8696 | 0.8308 | 0.8932 | **0.9838** | 0.9657 | 0.7753 | 0.9590 | 0.8548 | 0.7828 | 0.5290 |
| | xlm-roberta-large | 0.9599 | 0.9733 | 0.9850 | 0.9696 | **0.9946** | 0.9691 | 0.9816 | 0.9829 | 0.9733 | 0.9833 | 0.9365 |
| **llama-65b** | bert-base-multilingual-cased | 0.7821 | 0.8645 | 0.9288 | 0.7867 | 0.9076 | **0.9879** | 0.7835 | 0.9449 | 0.9782 | 0.9781 | 0.6580 |
| | electra-large-discriminator | 0.7392 | 0.9281 | 0.9320 | 0.9455 | 0.8964 | **0.9862** | 0.8995 | 0.9570 | 0.9189 | 0.9443 | 0.6784 |
| | gpt2-medium | 0.6580 | 0.9632 | 0.9491 | 0.9490 | 0.9016 | **0.9845** | 0.9279 | 0.9605 | 0.9765 | 0.9697 | 0.6849 |
| | mGPT | 0.8919 | 0.8853 | 0.9509 | 0.9058 | 0.9293 | **0.9862** | 0.7192 | 0.9691 | 0.9799 | 0.9731 | 0.5337 |
| | mdeberta-v3-base | 0.8006 | 0.8887 | 0.9831 | 0.8061 | 0.9004 | 0.9828 | 0.6450 | 0.9535 | **0.9899** | 0.9663 | 0.5974 |
| | roberta-large-openai-detector | 0.6731 | 0.9766 | 0.9763 | 0.8597 | 0.9239 | **0.9862** | 0.9598 | 0.9690 | 0.9714 | 0.9781 | 0.4521 |
| | xlm-roberta-large | 0.7446 | 0.8013 | 0.9373 | 0.8827 | 0.8984 | **0.9914** | 0.4673 | 0.9448 | 0.9899 | 0.9798 | 0.5992 |
| **opt-66b** | bert-base-multilingual-cased | 0.7967 | 0.7219 | 0.7674 | 0.7282 | 0.8716 | 0.8097 | 0.6880 | 0.8256 | **0.9168** | 0.9000 | 0.7730 |
| | electra-large-discriminator | 0.4285 | 0.7333 | 0.6311 | 0.7610 | **0.8950** | 0.7945 | 0.7222 | 0.8359 | 0.3401 | 0.3394 | 0.3731 |
| | gpt2-medium | 0.6023 | 0.8078 | 0.6790 | 0.7160 | 0.8877 | 0.7941 | 0.6774 | 0.7390 | **0.9070** | 0.8897 | 0.5969 |
| | mGPT | 0.9069 | 0.8440 | 0.8010 | 0.8186 | 0.9043 | 0.8626 | 0.7591 | 0.8012 | **0.9747** | 0.9729 | 0.6929 |
| | mdeberta-v3-base | 0.8913 | 0.9396 | **1.0000** | 0.8978 | 0.8479 | 0.8847 | 0.8763 | 0.8646 | 0.9882 | 0.9831 | 0.8073 |
| | roberta-large-openai-detector | 0.6401 | 0.6683 | 0.8647 | 0.7966 | **0.9097** | 0.8645 | 0.8275 | 0.8683 | 0.8985 | 0.8812 | 0.6668 |
| | xlm-roberta-large | 0.9410 | 0.9514 | **0.9967** | 0.8348 | 0.8879 | 0.8695 | 0.8959 | 0.9212 | 0.9746 | 0.9966 | 0.7806 |
| **opt-iml-max-1.3b** | bert-base-multilingual-cased | 0.9503 | 0.9327 | 0.9598 | 0.8388 | 0.9206 | 0.9219 | 0.9006 | 0.9224 | **0.9863** | 0.9845 | 0.9494 |
| | electra-large-discriminator | 0.3386 | 0.9243 | 0.8798 | 0.8805 | **0.9801** | 0.9672 | 0.9381 | 0.9378 | 0.3425 | 0.3429 | 0.4383 |
| | gpt2-medium | 0.6676 | **0.9815** | 0.9295 | 0.8375 | 0.9548 | 0.9585 | 0.9159 | 0.8588 | 0.9281 | 0.9255 | 0.6572 |
| | mGPT | 0.9760 | 0.9597 | 0.9243 | 0.9524 | 0.9567 | 0.9793 | 0.9565 | 0.9707 | **0.9983** | 0.9966 | 0.7681 |
| | mdeberta-v3-base | 0.9760 | 0.9882 | **1.0000** | 0.9575 | 0.9205 | 0.9672 | 0.9732 | 0.9810 | **1.0000** | 0.9966 | 0.7999 |
| | roberta-large-openai-detector | 0.6580 | 0.9511 | 0.9783 | 0.9405 | 0.9675 | **0.9810** | 0.9799 | 0.9326 | 0.9215 | 0.9191 | 0.6601 |
| | xlm-roberta-large | 0.9863 | 0.9966 | **1.0000** | 0.9932 | 0.9054 | 0.9689 | 0.9916 | 0.9707 | **1.0000** | 0.9983 | 0.8078 |
| **text-davinci-003** | bert-base-multilingual-cased | 0.8246 | 0.9583 | 0.9349 | 0.9527 | 0.9639 | **0.9640** | 0.9616 | 0.9351 | 0.9098 | 0.8775 | 0.9183 |
| | electra-large-discriminator | 0.3330 | 0.7572 | 0.3333 | 0.3377 | **0.9838** | 0.9623 | 0.5772 | 0.8433 | 0.3337 | 0.3330 | 0.3781 |
| | gpt2-medium | 0.7628 | 0.9299 | 0.6175 | 0.7370 | **0.9819** | 0.9365 | 0.7891 | 0.8164 | 0.8564 | 0.8406 | 0.4081 |
| | mGPT | 0.8462 | 0.6847 | 0.3333 | 0.9730 | **0.9801** | 0.9743 | 0.9599 | 0.9438 | 0.9499 | 0.8396 | 0.8442 |
| | mdeberta-v3-base | 0.7598 | 0.8536 | **0.9516** | 0.9118 | 0.8964 | 0.8446 | 0.8755 | 0.8891 | 0.9349 | 0.9244 | 0.9331 |
| | roberta-large-openai-detector | 0.6837 | 0.7831 | 0.8645 | 0.8792 | 0.8488 | **0.9257** | 0.8654 | 0.8806 | 0.8556 | 0.8590 | 0.6139 |
| | xlm-roberta-large | 0.6706 | 0.7938 | 0.8870 | 0.9357 | **0.9693** | 0.9298 | 0.9649 | 0.9062 | 0.9041 | 0.8336 | 0.6714 |
| **vicuna-13b** | bert-base-multilingual-cased | 0.8294 | 0.9533 | 0.8963 | 0.8902 | **0.9603** | 0.9486 | 0.9599 | 0.9420 | 0.9398 | 0.8928 | 0.7173 |
| | electra-large-discriminator | 0.3548 | 0.8538 | 0.4133 | 0.4349 | **0.9874** | 0.9536 | 0.6708 | 0.9148 | 0.3703 | 0.3400 | 0.4983 |
| | gpt2-medium | 0.7526 | 0.9010 | 0.6593 | 0.6846 | **0.9838** | 0.9331 | 0.7428 | 0.8293 | 0.8517 | 0.6033 | 0.4810 |
| | mGPT | 0.8467 | 0.8620 | 0.5100 | 0.9031 | **0.9801** | 0.9398 | 0.8907 | 0.9352 | 0.9413 | 0.8451 | 0.6522 |
| | mdeberta-v3-base | 0.8543 | 0.8078 | 0.8991 | 0.9204 | 0.8907 | 0.8410 | 0.8596 | 0.8721 | 0.9363 | **0.9364** | 0.7012 |
| | roberta-large-openai-detector | 0.7635 | 0.9214 | 0.8653 | 0.9070 | **0.9729** | 0.9570 | 0.8976 | 0.9281 | 0.8671 | 0.7857 | 0.7200 |
| | xlm-roberta-large | 0.8809 | 0.9517 | 0.9348 | 0.9594 | **0.9819** | 0.9709 | 0.9179 | 0.9213 | 0.9749 | 0.9565 | 0.7442 |
| | **All Detectors Mean** | 0.7463 | 0.8765 | 0.7918 | 0.8300 | 0.9472 | 0.9375 | 0.8407 | 0.9099 | 0.8592 | 0.8296 | 0.6558 |
| | **Multilingual Base Models Mean** | 0.8537 | 0.8977 | 0.8604 | 0.9073 | 0.9420 | 0.9372 | 0.8808 | 0.9253 | 0.9560 | 0.9374 | 0.7659 |
| | **Monolingual Base Models Mean** | 0.6031 | 0.8482 | 0.7003 | 0.7269 | 0.9542 | 0.9380 | 0.7872 | 0.8895 | 0.7301 | 0.6859 | 0.5090 |

Table 14: Cross-lingual performance of all detection models fine-tuned on all three train languages (evaluated per LLM).

| Train & Test LLM | Detector Base Model | Test Language | | | | | | | | | | |
|---|---|---|---|---|---|---|---|---|---|---|---|---|
| | | ar | ca | cs | de | en | es | nl | pt | ru | uk | zh |
| **alpaca-lora-30b** | bert-base-multilingual-cased | 0.4951 | 0.8743 | 0.8600 | 0.9053 | **0.9838** | 0.7617 | 0.9079 | 0.8579 | 0.8461 | 0.8144 | 0.5719 |
| | electra-large-discriminator | 0.3770 | 0.4539 | 0.4389 | 0.3827 | **0.9892** | 0.5344 | 0.4379 | 0.4684 | 0.3363 | 0.3333 | 0.4711 |
| | gpt2-medium | 0.3656 | 0.4420 | 0.4544 | 0.4038 | **0.9928** | 0.4063 | 0.4297 | 0.3814 | 0.3854 | 0.4195 | 0.4141 |
| | mGPT | 0.2930 | 0.7623 | 0.4958 | 0.8850 | **0.9801** | 0.7360 | 0.8594 | 0.8655 | 0.8209 | 0.8043 | 0.4268 |
| | mdeberta-v3-base | 0.1764 | 0.6872 | 0.8096 | 0.8529 | **0.9385** | 0.6356 | 0.7888 | 0.7340 | 0.7332 | 0.6824 | 0.4177 |
| | roberta-large-openai-detector | 0.3448 | 0.4963 | 0.4613 | 0.4216 | **0.9711** | 0.3945 | 0.6572 | 0.3721 | 0.3571 | 0.4340 | 0.4346 |
| | xlm-roberta-large | 0.3306 | 0.8378 | 0.8782 | 0.9425 | **0.9874** | 0.6985 | 0.9007 | 0.7872 | 0.8294 | 0.8524 | 0.5984 |
| **gpt-3.5-turbo** | bert-base-multilingual-cased | 0.9449 | 0.9178 | 0.9383 | 0.9257 | **0.9819** | 0.8718 | 0.9499 | 0.9298 | 0.9100 | 0.8946 | 0.9083 |
| | electra-large-discriminator | 0.4179 | 0.4001 | 0.3407 | 0.3524 | **0.9856** | 0.6747 | 0.4557 | 0.5561 | 0.4058 | 0.3972 | 0.3132 |
| | gpt2-medium | 0.3690 | 0.3407 | 0.3370 | 0.3450 | **0.9910** | 0.3301 | 0.3432 | 0.3313 | 0.3658 | 0.3539 | 0.3333 |
| | mGPT | 0.6786 | 0.9083 | 0.3333 | 0.9049 | **0.9874** | 0.8874 | 0.9633 | 0.9284 | 0.9232 | 0.8860 | 0.5494 |
| | mdeberta-v3-base | 0.8148 | 0.7353 | 0.8158 | 0.8783 | **0.9258** | 0.7020 | 0.8267 | 0.7834 | 0.6769 | 0.7348 | 0.7295 |
| | roberta-large-openai-detector | 0.6747 | 0.9182 | 0.8488 | 0.8416 | **0.9693** | 0.7811 | 0.8637 | 0.7284 | 0.6661 | 0.3600 | 0.5888 |
| | xlm-roberta-large | 0.9199 | 0.9800 | 0.9750 | 0.9408 | **0.9838** | 0.9192 | 0.9549 | 0.9385 | 0.9517 | 0.9464 | 0.7963 |
| **gpt-4** | bert-base-multilingual-cased | 0.8350 | 0.9009 | 0.9200 | 0.9307 | **0.9964** | 0.7323 | 0.9382 | 0.8389 | 0.8766 | 0.8662 | 0.8079 |
| | electra-large-discriminator | 0.4747 | 0.6222 | 0.3363 | 0.4691 | **0.9910** | 0.8320 | 0.5694 | 0.7838 | 0.3804 | 0.3378 | 0.3370 |
| | gpt2-medium | 0.3759 | 0.3516 | 0.3370 | 0.3413 | **0.9982** | 0.3553 | 0.3540 | 0.3565 | 0.3725 | 0.3468 | 0.3407 |
| | mGPT | 0.8481 | 0.9299 | 0.3333 | 0.9104 | **0.9928** | 0.7951 | 0.9314 | 0.9072 | 0.8798 | 0.8067 | 0.6570 |
| | mdeberta-v3-base | 0.8199 | 0.7249 | **0.9076** | 0.8679 | 0.6291 | 0.6506 | 0.8023 | 0.7064 | 0.7794 | 0.8271 | 0.6940 |
| | roberta-large-openai-detector | 0.5115 | 0.6923 | 0.5428 | 0.6090 | **0.9892** | 0.4870 | 0.7771 | 0.4381 | 0.4911 | 0.4030 | 0.6774 |
| | xlm-roberta-large | 0.8780 | 0.9093 | 0.9566 | 0.9645 | **0.9946** | 0.8286 | 0.9248 | 0.8808 | 0.9066 | 0.9298 | 0.8614 |
| **llama-65b** | bert-base-multilingual-cased | 0.6552 | 0.6417 | 0.7971 | 0.7019 | **0.9219** | 0.8057 | 0.6370 | 0.6880 | 0.7016 | 0.8031 | 0.5659 |
| | electra-large-discriminator | 0.3311 | 0.3363 | 0.3336 | 0.3536 | **0.9365** | 0.3537 | 0.3330 | 0.3311 | 0.3311 | 0.3326 | 0.3318 |
| | gpt2-medium | 0.3472 | 0.3979 | 0.5536 | 0.4564 | **0.9384** | 0.5798 | 0.3912 | 0.4192 | 0.3844 | 0.3912 | 0.4145 |
| | mGPT | 0.3369 | 0.3616 | 0.7761 | 0.7157 | **0.9147** | 0.5517 | 0.4591 | 0.5616 | 0.4846 | 0.5137 | 0.5556 |
| | mdeberta-v3-base | 0.7477 | 0.6646 | **0.9272** | 0.7560 | 0.8959 | 0.8662 | 0.6821 | 0.7176 | 0.7734 | 0.8533 | 0.5902 |
| | roberta-large-openai-detector | 0.3311 | 0.3607 | 0.3545 | 0.3408 | **0.9457** | 0.3460 | 0.3761 | 0.3395 | 0.3355 | 0.3529 | 0.3325 |
| | xlm-roberta-large | 0.4532 | 0.4074 | 0.8414 | 0.7198 | **0.9384** | 0.6397 | 0.4080 | 0.5535 | 0.6371 | 0.6560 | 0.4488 |
| **opt-66b** | bert-base-multilingual-cased | 0.4121 | 0.5241 | 0.6850 | 0.6789 | **0.9079** | 0.5558 | 0.5274 | 0.6377 | 0.6104 | 0.6477 | 0.5323 |
| | electra-large-discriminator | 0.3969 | 0.6936 | 0.6463 | 0.6972 | **0.9314** | 0.6785 | 0.6637 | 0.7006 | 0.3390 | 0.3303 | 0.3311 |
| | gpt2-medium | 0.4448 | 0.4992 | 0.4264 | 0.5941 | **0.8898** | 0.5432 | 0.4274 | 0.6423 | 0.4870 | 0.5507 | 0.4764 |
| | mGPT | 0.1983 | 0.5890 | 0.5708 | 0.6564 | **0.9185** | 0.5605 | 0.5826 | 0.6449 | 0.4463 | 0.4981 | 0.4464 |
| | mdeberta-v3-base | 0.2612 | 0.4316 | 0.5316 | 0.6257 | **0.7720** | 0.4713 | 0.4184 | 0.3897 | 0.5503 | 0.4716 | 0.5307 |
| | roberta-large-openai-detector | 0.3322 | 0.6773 | 0.4695 | 0.4529 | **0.9332** | 0.4155 | 0.6678 | 0.4105 | 0.3346 | 0.3886 | 0.3376 |
| | xlm-roberta-large | 0.4182 | 0.4089 | 0.6021 | 0.6750 | **0.9205** | 0.5871 | 0.4205 | 0.5935 | 0.5884 | 0.6220 | 0.5175 |
| **opt-iml-max-1.3b** | bert-base-multilingual-cased | 0.5705 | 0.6372 | 0.7086 | 0.7003 | **0.9549** | 0.6327 | 0.6705 | 0.6447 | 0.5501 | 0.6380 | 0.5188 |
| | electra-large-discriminator | 0.3280 | 0.6391 | 0.6211 | 0.7352 | **0.9964** | 0.8875 | 0.8502 | 0.8454 | 0.3280 | 0.3275 | 0.3292 |
| | gpt2-medium | 0.3740 | 0.8301 | 0.5994 | 0.6433 | **0.9801** | 0.7471 | 0.6005 | 0.6545 | 0.3770 | 0.3929 | 0.4977 |
| | mGPT | 0.2607 | 0.5663 | 0.5731 | 0.6369 | **0.9729** | 0.5881 | 0.5901 | 0.4952 | 0.2563 | 0.2537 | 0.4156 |
| | mdeberta-v3-base | 0.7309 | 0.8618 | 0.9212 | 0.8152 | **0.9729** | 0.7726 | 0.8558 | 0.7391 | 0.7542 | 0.8477 | 0.7328 |
| | roberta-large-openai-detector | 0.3280 | 0.8886 | 0.7955 | 0.6782 | **0.9856** | 0.7670 | 0.9075 | 0.6858 | 0.3429 | 0.4533 | 0.3737 |
| | xlm-roberta-large | 0.6694 | 0.7393 | 0.7423 | 0.8130 | **0.9711** | 0.7553 | 0.6195 | 0.6933 | 0.5689 | 0.5977 | 0.6165 |
| **text-davinci-003** | bert-base-multilingual-cased | 0.7853 | 0.9064 | 0.8815 | 0.8951 | **0.9693** | 0.7987 | 0.9115 | 0.8632 | 0.7773 | 0.7333 | 0.8883 |
| | electra-large-discriminator | 0.3985 | 0.3333 | 0.3333 | 0.3303 | **0.9910** | 0.3309 | 0.3330 | 0.3284 | 0.3741 | 0.4089 | 0.3333 |
| | gpt2-medium | 0.3330 | 0.3333 | 0.3333 | 0.3303 | **0.9892** | 0.3264 | 0.3330 | 0.3276 | 0.3283 | 0.3334 | 0.3326 |
| | mGPT | 0.6136 | 0.8932 | 0.3333 | 0.8701 | **0.9892** | 0.8484 | 0.9382 | 0.9079 | 0.6940 | 0.5687 | 0.5904 |
| | mdeberta-v3-base | 0.5791 | 0.7680 | 0.8940 | 0.8895 | **0.9074** | 0.6401 | 0.8748 | 0.7725 | 0.7341 | 0.6130 | 0.7199 |
| | roberta-large-openai-detector | 0.5757 | 0.7601 | 0.8132 | 0.8724 | **0.9946** | 0.7930 | 0.7339 | 0.7721 | 0.6125 | 0.3726 | 0.6949 |
| | xlm-roberta-large | 0.3899 | 0.9042 | 0.5268 | 0.9016 | **0.9856** | 0.8327 | 0.9482 | 0.8872 | 0.5981 | 0.3932 | 0.3987 |
| **vicuna-13b** | bert-base-multilingual-cased | 0.7679 | 0.8821 | 0.8366 | 0.8410 | **0.9874** | 0.7861 | 0.9382 | 0.8218 | 0.8335 | 0.8378 | 0.6379 |
| | electra-large-discriminator | 0.4056 | 0.5004 | 0.5494 | 0.5233 | **0.9838** | 0.7749 | 0.6136 | 0.6435 | 0.4744 | 0.4347 | 0.5352 |
| | gpt2-medium | 0.5084 | 0.3728 | 0.4513 | 0.4422 | **0.9856** | 0.5856 | 0.3776 | 0.4527 | 0.4134 | 0.4440 | 0.3841 |
| | mGPT | 0.3881 | 0.9079 | 0.4889 | 0.8254 | **0.9910** | 0.6963 | 0.9161 | 0.8043 | 0.6361 | 0.7215 | 0.6778 |
| | mdeberta-v3-base | 0.2578 | 0.7861 | 0.6983 | 0.8225 | 0.7928 | 0.6142 | **0.8809** | 0.7064 | 0.6031 | 0.6765 | 0.5488 |
| | roberta-large-openai-detector | 0.3330 | 0.4906 | 0.4113 | 0.4794 | **0.9511** | 0.4425 | 0.6971 | 0.3821 | 0.3326 | 0.3341 | 0.3488 |
| | xlm-roberta-large | 0.4062 | 0.9009 | 0.7648 | 0.9324 | **0.9892** | 0.6761 | 0.8613 | 0.7444 | 0.7618 | 0.7866 | 0.7595 |
| | **All Detectors Mean** | 0.4931 | 0.6568 | 0.6270 | 0.6871 | 0.9529 | 0.6476 | 0.6800 | 0.6497 | 0.5759 | 0.5716 | 0.5299 |
| | **Multilingual Base Models Mean** | 0.5605 | 0.7484 | 0.7289 | 0.8244 | 0.9392 | 0.7156 | 0.7778 | 0.7508 | 0.7092 | 0.7118 | 0.6160 |
| | **Monolingual Base Models Mean** | 0.4033 | 0.5346 | 0.4912 | 0.5040 | 0.9712 | 0.5570 | 0.5497 | 0.5150 | 0.3981 | 0.3847 | 0.4152 |

Table 15: Cross-lingual performance of all detection models fine-tuned on the English language with 3-times more train samples (evaluated per LLM).

| Train LLM | Detector Base Model | Test LLM | | | | | | | |
|---|---|---|---|---|---|---|---|---|---|
| | | text-davinci-003 | gpt-3.5-turbo | gpt-4 | alpaca-lora-30b | vicuna-13b | llama-65b | opt-66b | opt-iml-max-1.3b |
| alpaca-lora-30b | bert-base-multilingual-cased | 0.8102 | **0.8237** | 0.7964 | 0.7889 | 0.7622 | 0.4336 | 0.4315 | 0.4863 |
| | electra-large-discriminator | 0.5356 | 0.5243 | 0.5001 | **0.5698** | 0.5460 | 0.4060 | 0.4595 | 0.5168 |
| | gpt2-medium | 0.5267 | 0.4589 | 0.4438 | 0.5426 | 0.4768 | **0.5626** | 0.4401 | 0.4939 |
| | mGPT | 0.7656 | **0.8022** | 0.7437 | 0.7556 | 0.7396 | 0.4427 | 0.4386 | 0.3624 |
| | mdeberta-v3-base | 0.7795 | 0.8118 | **0.8242** | 0.7428 | 0.7404 | 0.3676 | 0.3285 | 0.3175 |
| | roberta-large-openai-detector | **0.5663** | 0.5660 | 0.5646 | 0.5627 | 0.5556 | 0.4595 | 0.5272 | 0.5061 |
| | xlm-roberta-large | 0.7940 | **0.8264** | 0.8029 | 0.7643 | 0.7542 | 0.4912 | 0.4660 | 0.5107 |
| gpt-3.5-turbo | bert-base-multilingual-cased | 0.8512 | **0.9106** | 0.8841 | 0.7797 | 0.8281 | 0.3758 | 0.4023 | 0.3959 |
| | electra-large-discriminator | 0.5937 | 0.6052 | 0.5879 | 0.6016 | **0.6087** | 0.4381 | 0.4511 | 0.4864 |
| | gpt2-medium | 0.4147 | 0.4302 | 0.4382 | 0.4215 | 0.4532 | **0.4995** | 0.3672 | 0.4162 |
| | mGPT | 0.7572 | **0.8286** | 0.7858 | 0.6841 | 0.7503 | 0.3726 | 0.3879 | 0.3457 |
| | mdeberta-v3-base | 0.6966 | 0.7136 | **0.7313** | 0.6586 | 0.6704 | 0.3248 | 0.2977 | 0.2729 |
| | roberta-large-openai-detector | **0.6167** | 0.6141 | 0.6105 | 0.6013 | 0.5940 | 0.4211 | 0.5461 | 0.4714 |
| | xlm-roberta-large | 0.7198 | **0.8308** | 0.7862 | 0.6829 | 0.7215 | 0.3967 | 0.4055 | 0.3881 |
| gpt-4 | bert-base-multilingual-cased | 0.7090 | 0.7954 | **0.8458** | 0.6428 | 0.6632 | 0.3744 | 0.3749 | 0.3585 |
| | electra-large-discriminator | 0.4972 | 0.5225 | **0.5362** | 0.4759 | 0.5156 | 0.3720 | 0.3757 | 0.3755 |
| | gpt2-medium | 0.4037 | 0.4334 | 0.4460 | 0.4199 | 0.4387 | **0.5207** | 0.3628 | 0.4035 |
| | mGPT | 0.7058 | 0.7616 | **0.7657** | 0.6387 | 0.6981 | 0.4186 | 0.4255 | 0.3558 |
| | mdeberta-v3-base | 0.6201 | 0.8077 | **0.8469** | 0.5331 | 0.6150 | 0.3346 | 0.3385 | 0.3357 |
| | roberta-large-openai-detector | 0.6070 | 0.6131 | **0.6136** | 0.5804 | 0.5889 | 0.3577 | 0.5085 | 0.3777 |
| | xlm-roberta-large | 0.6526 | 0.7151 | **0.7742** | 0.5707 | 0.6754 | 0.3815 | 0.3821 | 0.3743 |
| llama-65b | bert-base-multilingual-cased | 0.2969 | 0.3174 | 0.4327 | 0.3828 | 0.4522 | **0.6841** | 0.6015 | 0.6403 |
| | electra-large-discriminator | 0.5096 | 0.4843 | 0.4806 | 0.4937 | 0.4921 | **0.5111** | 0.5108 | 0.5095 |
| | gpt2-medium | 0.4531 | 0.4221 | 0.4092 | 0.4193 | 0.4644 | **0.5337** | 0.4957 | 0.4954 |
| | mGPT | 0.4849 | 0.4824 | 0.5086 | 0.4783 | 0.5308 | **0.5608** | 0.5337 | 0.5410 |
| | mdeberta-v3-base | 0.5439 | 0.5356 | 0.5626 | 0.5473 | 0.6054 | **0.6348** | 0.6075 | 0.6318 |
| | roberta-large-openai-detector | **0.4157** | 0.4153 | 0.4157 | 0.4131 | 0.4133 | 0.4124 | 0.4099 | 0.4060 |
| | xlm-roberta-large | 0.4817 | 0.4691 | 0.5183 | 0.5133 | 0.5757 | **0.6625** | 0.6145 | 0.6553 |
| opt-66b | bert-base-multilingual-cased | 0.4144 | 0.4443 | 0.5264 | 0.4583 | 0.5037 | **0.6291** | 0.5623 | 0.5766 |
| | electra-large-discriminator | 0.4734 | 0.4710 | 0.5217 | 0.5028 | 0.5515 | **0.6779** | 0.5974 | 0.6285 |
| | gpt2-medium | 0.5210 | 0.4556 | 0.4574 | 0.4769 | 0.4977 | **0.6650** | 0.5379 | 0.5676 |
| | mGPT | 0.5835 | 0.6185 | **0.6248** | 0.4870 | 0.6169 | 0.6016 | 0.5344 | 0.4388 |
| | mdeberta-v3-base | 0.4586 | 0.5203 | 0.5924 | 0.4266 | 0.6025 | **0.6556** | 0.5969 | 0.5744 |
| | roberta-large-openai-detector | 0.4785 | 0.4776 | **0.4787** | 0.4731 | 0.4726 | 0.4619 | 0.4708 | 0.4719 |
| | xlm-roberta-large | 0.5180 | 0.5048 | 0.5545 | 0.4707 | 0.5929 | **0.7085** | 0.6116 | 0.6762 |
| opt-iml-max-1.3b | bert-base-multilingual-cased | 0.5108 | 0.5289 | 0.5330 | 0.5173 | 0.5442 | **0.5635** | 0.5450 | 0.5626 |
| | electra-large-discriminator | 0.4745 | 0.4613 | 0.4788 | 0.5011 | 0.5009 | 0.6021 | 0.5657 | **0.6240** |
| | gpt2-medium | 0.5220 | 0.4936 | 0.5053 | 0.5192 | 0.5431 | **0.6894** | 0.5497 | 0.6141 |
| | mGPT | 0.6135 | 0.6200 | 0.6061 | 0.6004 | **0.6331** | 0.6238 | 0.5904 | 0.6172 |
| | mdeberta-v3-base | 0.5972 | 0.5374 | 0.5093 | 0.6664 | 0.7953 | 0.8235 | 0.8139 | **0.9138** |
| | roberta-large-openai-detector | 0.6543 | 0.6591 | 0.6491 | 0.6438 | 0.6535 | 0.5722 | 0.6484 | **0.6772** |
| | xlm-roberta-large | 0.5576 | 0.5218 | 0.5196 | 0.5736 | 0.6675 | 0.7572 | 0.6830 | **0.7731** |
| text-davinci-003 | bert-base-multilingual-cased | 0.8289 | **0.8692** | 0.8116 | 0.7427 | 0.7640 | 0.3490 | 0.3876 | 0.3703 |
| | electra-large-discriminator | 0.5878 | 0.5924 | 0.5310 | 0.5775 | **0.5942** | 0.4013 | 0.4614 | 0.5302 |
| | gpt2-medium | 0.4172 | 0.4284 | 0.4378 | 0.4176 | 0.4411 | **0.4916** | 0.3651 | 0.4088 |
| | mGPT | 0.7636 | **0.8044** | 0.7709 | 0.6834 | 0.7306 | 0.3624 | 0.4103 | 0.3518 |
| | mdeberta-v3-base | 0.7193 | 0.7398 | **0.7533** | 0.6600 | 0.6873 | 0.3190 | 0.3248 | 0.2649 |
| | roberta-large-openai-detector | **0.5722** | 0.5691 | 0.5671 | 0.5618 | 0.5534 | 0.4024 | 0.5216 | 0.4817 |
| | xlm-roberta-large | 0.7430 | **0.7991** | 0.7469 | 0.6847 | 0.7062 | 0.4158 | 0.4656 | 0.5216 |
| vicuna-13b | bert-base-multilingual-cased | 0.8212 | **0.8745** | 0.8672 | 0.7552 | 0.8353 | 0.4419 | 0.4167 | 0.4256 |
| | electra-large-discriminator | 0.5655 | 0.5769 | 0.5669 | 0.5823 | 0.5920 | 0.5197 | 0.5161 | **0.6136** |
| | gpt2-medium | **0.5558** | 0.4521 | 0.4421 | 0.5117 | 0.5204 | 0.5360 | 0.4994 | 0.5329 |
| | mGPT | 0.7242 | **0.7568** | 0.7366 | 0.6741 | 0.7312 | 0.4275 | 0.4706 | 0.3665 |
| | mdeberta-v3-base | 0.5538 | 0.5773 | **0.5925** | 0.5141 | 0.5328 | 0.4013 | 0.2795 | 0.2131 |
| | roberta-large-openai-detector | **0.6170** | 0.6150 | 0.6093 | 0.6029 | 0.5980 | 0.4421 | 0.5623 | 0.5310 |
| | xlm-roberta-large | 0.7038 | **0.7807** | 0.7476 | 0.6598 | 0.7234 | 0.4878 | 0.4896 | 0.5116 |

Table 16: Cross-generator performance of all detection models fine-tuned on the English language (evaluated per LLM).

| Train LLM | Detector Base Model | Test LLM | | | | | | | |
|---|---|---|---|---|---|---|---|---|---|
| | | text-davinci-003 | gpt-3.5-turbo | gpt-4 | alpaca-lora-30b | vicuna-13b | llama-65b | opt-66b | opt-iml-max-1.3b |
| alpaca-lora-30b | bert-base-multilingual-cased | 0.8581 | **0.8905** | 0.8078 | 0.8295 | 0.8268 | 0.5157 | 0.4523 | 0.5286 |
| | electra-large-discriminator | 0.5473 | 0.5414 | 0.5105 | 0.5562 | **0.5624** | 0.5363 | 0.4681 | 0.4795 |
| | gpt2-medium | 0.6630 | 0.5975 | 0.4491 | **0.7259** | 0.5730 | 0.5714 | 0.4916 | 0.5837 |
| | mGPT | 0.7748 | 0.8005 | 0.6966 | **0.8025** | 0.7508 | 0.6114 | 0.5314 | 0.5592 |
| | mdeberta-v3-base | **0.8632** | 0.8505 | 0.8237 | 0.8465 | 0.8484 | 0.4766 | 0.5173 | 0.6004 |
| | roberta-large-openai-detector | 0.6667 | **0.6847** | 0.6452 | 0.6807 | 0.6629 | 0.5254 | 0.4764 | 0.3758 |
| | xlm-roberta-large | 0.8169 | 0.8136 | 0.6452 | **0.8599** | 0.7653 | 0.4279 | 0.4353 | 0.5307 |
| gpt-3.5-turbo | bert-base-multilingual-cased | 0.8364 | **0.9011** | 0.8290 | 0.7524 | 0.7909 | 0.3473 | 0.3842 | 0.3736 |
| | electra-large-discriminator | 0.5457 | **0.5958** | 0.5620 | 0.5250 | 0.5732 | 0.4545 | 0.4451 | 0.4277 |
| | gpt2-medium | 0.5610 | **0.5785** | 0.5022 | 0.5185 | 0.5366 | 0.5299 | 0.3966 | 0.5121 |
| | mGPT | 0.7977 | **0.8596** | 0.8033 | 0.7172 | 0.7721 | 0.3934 | 0.4093 | 0.3702 |
| | mdeberta-v3-base | 0.8451 | **0.9266** | 0.9040 | 0.7635 | 0.8030 | 0.3472 | 0.3405 | 0.3264 |
| | roberta-large-openai-detector | 0.6670 | **0.6981** | 0.6875 | 0.6202 | 0.6554 | 0.4468 | 0.4257 | 0.3329 |
| | xlm-roberta-large | 0.8127 | **0.8939** | 0.8010 | 0.7125 | 0.7738 | 0.3816 | 0.3816 | 0.4005 |
| gpt-4 | bert-base-multilingual-cased | 0.8240 | 0.8702 | **0.8802** | 0.7418 | 0.7969 | 0.4285 | 0.4251 | 0.3798 |
| | electra-large-discriminator | 0.5456 | 0.5537 | **0.5785** | 0.5060 | 0.5510 | 0.4553 | 0.4632 | 0.4512 |
| | gpt2-medium | 0.4109 | 0.5059 | **0.5551** | 0.4236 | 0.4680 | 0.5172 | 0.3713 | 0.4303 |
| | mGPT | 0.7428 | 0.8420 | **0.8526** | 0.6301 | 0.7372 | 0.4196 | 0.3823 | 0.3614 |
| | mdeberta-v3-base | 0.7977 | 0.9095 | **0.9113** | 0.6895 | 0.7768 | 0.3573 | 0.3471 | 0.3371 |
| | roberta-large-openai-detector | 0.6422 | 0.6892 | **0.7063** | 0.5845 | 0.6375 | 0.4231 | 0.3862 | 0.3187 |
| | xlm-roberta-large | 0.6783 | 0.7627 | **0.7680** | 0.6233 | 0.7000 | 0.3695 | 0.3670 | 0.3783 |
| llama-65b | bert-base-multilingual-cased | 0.3458 | 0.3815 | 0.4299 | 0.4424 | 0.5081 | **0.8214** | 0.6687 | 0.7356 |
| | electra-large-discriminator | 0.4849 | 0.4632 | 0.4809 | 0.5045 | 0.5231 | **0.7180** | 0.5699 | 0.6151 |
| | gpt2-medium | 0.4832 | 0.4536 | 0.4721 | 0.4963 | 0.5024 | **0.7405** | 0.5325 | 0.5924 |
| | mGPT | 0.5262 | 0.5447 | 0.5199 | 0.5776 | 0.5742 | **0.6943** | 0.5542 | 0.6419 |
| | mdeberta-v3-base | 0.4746 | 0.4224 | 0.4382 | 0.5887 | 0.6442 | **0.8678** | 0.7715 | 0.8579 |
| | roberta-large-openai-detector | 0.6111 | 0.6331 | 0.6282 | 0.5793 | 0.6229 | **0.6602** | 0.5422 | 0.4934 |
| | xlm-roberta-large | 0.4259 | 0.4284 | 0.4451 | 0.5757 | 0.6168 | **0.8565** | 0.7240 | 0.8332 |
| opt-66b | bert-base-multilingual-cased | 0.5182 | 0.5420 | 0.5370 | 0.5378 | 0.6400 | **0.7476** | 0.7074 | 0.7397 |
| | electra-large-discriminator | 0.4604 | 0.4612 | 0.4785 | 0.4740 | 0.5282 | **0.6504** | 0.6009 | 0.6205 |
| | gpt2-medium | 0.5741 | 0.5033 | 0.4575 | 0.5371 | 0.5645 | 0.6102 | 0.6019 | **0.6506** |
| | mGPT | 0.5684 | 0.5752 | 0.5390 | 0.5698 | 0.5986 | 0.5700 | 0.5972 | **0.6038** |
| | mdeberta-v3-base | 0.6073 | 0.4019 | 0.4000 | 0.7470 | 0.7419 | 0.8027 | 0.8651 | **0.8861** |
| | roberta-large-openai-detector | 0.6509 | 0.6478 | 0.6341 | 0.6311 | 0.6360 | 0.6075 | **0.6683** | 0.6683 |
| | xlm-roberta-large | 0.5889 | 0.4781 | 0.4654 | 0.6340 | 0.6626 | 0.7066 | 0.7053 | **0.7086** |
| opt-iml-max-1.3b | bert-base-multilingual-cased | 0.3503 | 0.3595 | 0.3455 | 0.4341 | 0.5012 | 0.7651 | 0.6947 | **0.9178** |
| | electra-large-discriminator | 0.5284 | 0.5251 | 0.4909 | 0.5445 | 0.5738 | 0.6557 | 0.5849 | **0.7075** |
| | gpt2-medium | 0.5215 | 0.4668 | 0.4399 | 0.4763 | 0.5111 | 0.5719 | 0.4972 | **0.6770** |
| | mGPT | 0.4806 | 0.4060 | 0.3641 | 0.6005 | 0.5902 | 0.7384 | 0.6777 | **0.8787** |
| | mdeberta-v3-base | 0.4692 | 0.3456 | 0.3352 | 0.5788 | 0.5674 | 0.5690 | 0.8046 | **0.9323** |
| | roberta-large-openai-detector | 0.6303 | 0.6232 | 0.5829 | 0.6268 | 0.6424 | 0.5714 | 0.6498 | **0.7347** |
| | xlm-roberta-large | 0.3438 | 0.3310 | 0.3320 | 0.4737 | 0.4500 | 0.5210 | 0.6757 | **0.8856** |
| text-davinci-003 | bert-base-multilingual-cased | 0.8398 | **0.8700** | 0.7840 | 0.7585 | 0.8105 | 0.3832 | 0.4459 | 0.4992 |
| | electra-large-discriminator | **0.5908** | 0.5807 | 0.5429 | 0.5449 | 0.5745 | 0.4342 | 0.4713 | 0.4590 |
| | gpt2-medium | **0.7275** | 0.6582 | 0.4750 | 0.6549 | 0.6159 | 0.4904 | 0.4822 | 0.6132 |
| | mGPT | 0.8230 | **0.8604** | 0.7730 | 0.7351 | 0.7823 | 0.4025 | 0.4022 | 0.3777 |
| | mdeberta-v3-base | 0.8459 | **0.8898** | 0.8881 | 0.7783 | 0.8136 | 0.3620 | 0.3554 | 0.3517 |
| | roberta-large-openai-detector | 0.6641 | **0.6923** | 0.6602 | 0.6147 | 0.6459 | 0.4523 | 0.4150 | 0.3379 |
| | xlm-roberta-large | 0.9113 | **0.9345** | 0.8569 | 0.7909 | 0.8627 | 0.3898 | 0.4001 | 0.4358 |
| vicuna-13b | bert-base-multilingual-cased | 0.8525 | **0.8954** | 0.8444 | 0.7812 | 0.8483 | 0.4819 | 0.4452 | 0.4992 |
| | electra-large-discriminator | 0.5415 | 0.5950 | 0.5707 | 0.5428 | **0.6017** | 0.5586 | 0.4745 | 0.4876 |
| | gpt2-medium | **0.7084** | 0.6886 | 0.5819 | 0.6673 | 0.7066 | 0.5962 | 0.5040 | 0.6613 |
| | mGPT | 0.7824 | **0.7967** | 0.7664 | 0.7500 | 0.7815 | 0.6007 | 0.5111 | 0.4828 |
| | mdeberta-v3-base | 0.8518 | **0.9009** | 0.8630 | 0.7728 | 0.8656 | 0.4793 | 0.4083 | 0.4794 |
| | roberta-large-openai-detector | 0.6966 | **0.7304** | 0.6872 | 0.6489 | 0.7141 | 0.5363 | 0.4789 | 0.4041 |
| | xlm-roberta-large | 0.7742 | 0.7924 | 0.6991 | 0.7086 | **0.8133** | 0.5095 | 0.5060 | 0.6679 |

Table 17: Cross-generator performance of all detection models fine-tuned on the Spanish language (evaluated per LLM).

| Train LLM | Detector Base Model | Test LLM | | | | | | | |
|---|---|---|---|---|---|---|---|---|---|
| | | text-davinci-003 | gpt-3.5-turbo | gpt-4 | alpaca-lora-30b | vicuna-13b | llama-65b | opt-66b | opt-iml-max-1.3b |
| alpaca-lora-30b | bert-base-multilingual-cased | 0.7279 | **0.7424** | 0.7276 | 0.7127 | 0.7079 | 0.4906 | 0.4891 | 0.5071 |
| | electra-large-discriminator | 0.4746 | 0.4737 | 0.4906 | 0.4889 | 0.4873 | **0.6388** | 0.5265 | 0.5725 |
| | gpt2-medium | 0.4154 | 0.4978 | 0.4338 | **0.6095** | 0.4154 | 0.5750 | 0.3855 | 0.4667 |
| | mGPT | 0.6885 | **0.7089** | 0.6610 | 0.7077 | 0.6784 | 0.5698 | 0.5004 | 0.4986 |
| | mdeberta-v3-base | 0.8700 | 0.8366 | 0.7959 | **0.9018** | 0.8637 | 0.5788 | 0.7436 | 0.8611 |
| | roberta-large-openai-detector | 0.5134 | 0.5240 | 0.5139 | **0.5359** | 0.4923 | 0.4890 | 0.4723 | 0.4254 |
| | xlm-roberta-large | 0.8739 | 0.8502 | 0.7762 | **0.9212** | 0.8470 | 0.5112 | 0.5750 | 0.8006 |
| gpt-3.5-turbo | bert-base-multilingual-cased | 0.8358 | **0.8834** | 0.8548 | 0.7617 | 0.8171 | 0.3808 | 0.4363 | 0.4259 |
| | electra-large-discriminator | 0.4347 | 0.4889 | **0.4983** | 0.4408 | 0.4384 | 0.4084 | 0.3857 | 0.3908 |
| | gpt2-medium | 0.4234 | **0.4879** | 0.4599 | 0.4539 | 0.4328 | 0.4395 | 0.3623 | 0.4325 |
| | mGPT | 0.7712 | **0.8387** | 0.7840 | 0.7107 | 0.7594 | 0.3828 | 0.4280 | 0.3666 |
| | mdeberta-v3-base | 0.8608 | **0.9185** | 0.9074 | 0.8127 | 0.8515 | 0.4025 | 0.3799 | 0.3644 |
| | roberta-large-openai-detector | 0.4998 | **0.5257** | 0.5247 | 0.5126 | 0.4993 | 0.4612 | 0.4788 | 0.3629 |
| | xlm-roberta-large | 0.9057 | **0.9472** | 0.9320 | 0.8523 | 0.8832 | 0.4311 | 0.4196 | 0.4130 |
| gpt-4 | bert-base-multilingual-cased | 0.7354 | 0.7671 | **0.7747** | 0.6645 | 0.7139 | 0.4094 | 0.4649 | 0.4051 |
| | electra-large-discriminator | 0.4297 | 0.4782 | 0.4788 | 0.4560 | 0.4521 | **0.4829** | 0.4246 | 0.4328 |
| | gpt2-medium | 0.4549 | 0.4935 | **0.5003** | 0.4893 | 0.4583 | 0.4802 | 0.4106 | 0.4643 |
| | mGPT | 0.7582 | 0.8458 | **0.8566** | 0.6444 | 0.7497 | 0.3996 | 0.4090 | 0.3656 |
| | mdeberta-v3-base | 0.7805 | 0.8126 | **0.8134** | 0.7199 | 0.7597 | 0.3770 | 0.3834 | 0.3425 |
| | roberta-large-openai-detector | 0.4814 | 0.4983 | **0.5355** | 0.4818 | 0.4794 | 0.4322 | 0.4600 | 0.3530 |
| | xlm-roberta-large | 0.8743 | 0.9378 | **0.9499** | 0.7668 | 0.8342 | 0.3855 | 0.3779 | 0.3651 |
| llama-65b | bert-base-multilingual-cased | 0.4444 | 0.4964 | 0.5551 | 0.4455 | 0.5476 | **0.7071** | 0.6247 | 0.6517 |
| | electra-large-discriminator | 0.4662 | 0.4574 | 0.4546 | 0.4690 | 0.4708 | **0.5003** | 0.4754 | 0.4648 |
| | gpt2-medium | 0.3999 | 0.4182 | 0.4093 | 0.4423 | 0.4377 | **0.6506** | 0.4791 | 0.4855 |
| | mGPT | 0.4844 | 0.4835 | 0.4891 | 0.5283 | 0.5146 | **0.5925** | 0.5171 | 0.5794 |
| | mdeberta-v3-base | 0.5546 | 0.4236 | 0.4816 | 0.6273 | 0.6372 | **0.7263** | 0.7123 | 0.7260 |
| | roberta-large-openai-detector | 0.4379 | 0.4328 | 0.4487 | 0.4540 | 0.4430 | **0.5352** | 0.4778 | 0.4396 |
| | xlm-roberta-large | 0.4767 | 0.4478 | 0.4730 | 0.5810 | 0.5973 | **0.7443** | 0.7012 | 0.7417 |
| opt-66b | bert-base-multilingual-cased | 0.3359 | 0.3343 | 0.3592 | 0.3989 | 0.4029 | **0.7943** | 0.7091 | 0.7573 |
| | electra-large-discriminator | 0.4473 | 0.4202 | 0.4189 | 0.4424 | 0.4409 | 0.4619 | **0.4759** | 0.4674 |
| | gpt2-medium | 0.4902 | 0.4222 | 0.3950 | 0.4765 | 0.4634 | **0.5932** | 0.5526 | 0.5827 |
| | mGPT | 0.3959 | 0.3604 | 0.3596 | 0.5127 | 0.4866 | 0.6481 | 0.6793 | **0.8315** |
| | mdeberta-v3-base | 0.4262 | 0.3398 | 0.3375 | 0.5127 | 0.4807 | 0.4615 | 0.7289 | **0.8663** |
| | roberta-large-openai-detector | 0.4435 | 0.4209 | 0.4359 | 0.4382 | 0.4294 | 0.4511 | **0.4890** | 0.4148 |
| | xlm-roberta-large | 0.3934 | 0.3304 | 0.3320 | 0.5074 | 0.4554 | 0.6092 | 0.7948 | **0.9364** |
| opt-iml-max-1.3b | bert-base-multilingual-cased | 0.3321 | 0.3321 | 0.3327 | 0.3456 | 0.3372 | 0.3776 | 0.4305 | **0.7076** |
| | electra-large-discriminator | 0.5100 | 0.4353 | 0.3585 | 0.5607 | 0.4552 | 0.5110 | 0.5510 | **0.5909** |
| | gpt2-medium | 0.4438 | 0.3922 | 0.3737 | 0.4431 | 0.4279 | 0.5229 | 0.4935 | **0.5983** |
| | mGPT | 0.3341 | 0.3257 | 0.3381 | 0.3707 | 0.3652 | 0.4328 | 0.4766 | **0.7054** |
| | mdeberta-v3-base | 0.3482 | 0.3401 | 0.3371 | 0.4485 | 0.3843 | 0.3611 | 0.5402 | **0.7068** |
| | roberta-large-openai-detector | **0.5199** | 0.4835 | 0.4789 | 0.4998 | 0.4926 | 0.4773 | 0.5193 | 0.5197 |
| | xlm-roberta-large | 0.3360 | 0.3323 | 0.3312 | 0.4210 | 0.3627 | 0.3791 | 0.5594 | **0.7996** |
| text-davinci-003 | bert-base-multilingual-cased | 0.7808 | **0.7940** | 0.6553 | 0.7189 | 0.6841 | 0.3456 | 0.3798 | 0.3753 |
| | electra-large-discriminator | 0.4372 | 0.3887 | 0.3642 | 0.4289 | 0.4131 | **0.4951** | 0.4518 | 0.4553 |
| | gpt2-medium | **0.5147** | 0.3874 | 0.3753 | 0.4490 | 0.4148 | 0.4860 | 0.4456 | 0.5071 |
| | mGPT | **0.7682** | 0.7322 | 0.6571 | 0.7275 | 0.6993 | 0.3722 | 0.4151 | 0.3919 |
| | mdeberta-v3-base | 0.6726 | 0.5051 | 0.4342 | **0.7695** | 0.6230 | 0.3618 | 0.6033 | 0.6648 |
| | roberta-large-openai-detector | **0.5439** | 0.4846 | 0.4907 | 0.5025 | 0.4830 | 0.4156 | 0.4845 | 0.3512 |
| | xlm-roberta-large | 0.7099 | 0.5923 | 0.4626 | **0.7232** | 0.6210 | 0.3623 | 0.4792 | 0.6351 |
| vicuna-13b | bert-base-multilingual-cased | 0.5322 | **0.5444** | 0.5417 | 0.5321 | 0.5429 | 0.4956 | 0.4570 | 0.4338 |
| | electra-large-discriminator | 0.4613 | 0.4599 | 0.4605 | 0.4708 | 0.4680 | **0.4852** | 0.4774 | 0.4705 |
| | gpt2-medium | 0.4400 | 0.4965 | 0.4809 | 0.5031 | 0.5477 | **0.6459** | 0.4519 | 0.5139 |
| | mGPT | 0.6592 | 0.6760 | 0.6616 | 0.6673 | **0.6765** | 0.5875 | 0.5462 | 0.5849 |
| | mdeberta-v3-base | 0.8342 | 0.8001 | 0.7582 | 0.8513 | 0.8957 | 0.6075 | 0.7289 | **0.9080** |
| | roberta-large-openai-detector | 0.4998 | 0.5137 | 0.5146 | 0.5088 | **0.5269** | 0.5206 | 0.4873 | 0.4461 |
| | xlm-roberta-large | 0.8429 | 0.8318 | 0.7992 | 0.8220 | **0.8852** | 0.5332 | 0.5935 | 0.7620 |

Table 18: Cross-generator performance of all detection models fine-tuned on the Russian language (evaluated per LLM).

| Train Language | Train LLM | text-davinci-003 | gpt-3.5-turbo | gpt-4 | alpaca-lora-30b | vicuna-13b | llama-65b | opt-66b | opt-iml-max-1.3b |
|---|---|---|---|---|---|---|---|---|---|
| en | text-davinci-003 | 1.0000 | 0.9630 | 0.9091 | 0.9606 | 0.9153 | -0.5403 | -0.3397 | -0.3932 |
| | gpt-3.5-turbo | 0.9630 | 1.0000 | 0.9755 | 0.9000 | 0.8939 | -0.6074 | -0.4288 | -0.4835 |
| | gpt-4 | 0.9091 | 0.9755 | 1.0000 | 0.8362 | 0.8697 | -0.5739 | -0.4265 | -0.4911 |
| | alpaca-lora-30b | 0.9606 | 0.9000 | 0.8362 | 1.0000 | 0.9185 | -0.4377 | -0.2123 | -0.2412 |
| | vicuna-13b | 0.9153 | 0.8939 | 0.8697 | 0.9185 | 1.0000 | -0.2738 | -0.0814 | -0.1394 |
| | llama-65b | -0.5403 | -0.6074 | -0.5739 | -0.4377 | -0.2738 | 1.0000 | 0.8306 | 0.8684 |
| | opt-66b | -0.3397 | -0.4288 | -0.4265 | -0.2123 | -0.0814 | 0.8306 | 1.0000 | 0.9320 |
| | opt-iml-max-1.3b | -0.3932 | -0.4835 | -0.4911 | -0.2412 | -0.1394 | 0.8684 | 0.9320 | 1.0000 |
| es | text-davinci-003 | 1.0000 | 0.9559 | 0.8838 | 0.9107 | 0.9268 | -0.6977 | -0.6220 | -0.6146 |
| | gpt-3.5-turbo | 0.9559 | 1.0000 | 0.9642 | 0.8045 | 0.8793 | -0.7677 | -0.7582 | -0.7430 |
| | gpt-4 | 0.8838 | 0.9642 | 1.0000 | 0.7110 | 0.8475 | -0.7397 | -0.7464 | -0.7772 |
| | alpaca-lora-30b | 0.9107 | 0.8045 | 0.7110 | 1.0000 | 0.9211 | -0.4645 | -0.3431 | -0.3439 |
| | vicuna-13b | 0.9268 | 0.8793 | 0.8475 | 0.9211 | 1.0000 | -0.5080 | -0.4104 | -0.4309 |
| | llama-65b | -0.6977 | -0.7677 | -0.7397 | -0.4645 | -0.5080 | 1.0000 | 0.8401 | 0.7872 |
| | opt-66b | -0.6220 | -0.7582 | -0.7464 | -0.3431 | -0.4104 | 0.8401 | 1.0000 | 0.9164 |
| | opt-iml-max-1.3b | -0.6146 | -0.7430 | -0.7772 | -0.3439 | -0.4309 | 0.7872 | 0.9164 | 1.0000 |
| ru | text-davinci-003 | 1.0000 | 0.9575 | 0.9176 | 0.9459 | 0.9629 | -0.3001 | -0.1209 | -0.1660 |
| | gpt-3.5-turbo | 0.9575 | 1.0000 | 0.9806 | 0.8830 | 0.9405 | -0.3082 | -0.2675 | -0.2963 |
| | gpt-4 | 0.9176 | 0.9806 | 1.0000 | 0.8255 | 0.9281 | -0.2496 | -0.2466 | -0.3122 |
| | alpaca-lora-30b | 0.9459 | 0.8830 | 0.8255 | 1.0000 | 0.9460 | -0.1665 | 0.0646 | 0.0440 |
| | vicuna-13b | 0.9629 | 0.9405 | 0.9281 | 0.9460 | 1.0000 | -0.1229 | 0.0303 | -0.0324 |
| | llama-65b | -0.3001 | -0.3082 | -0.2496 | -0.1665 | -0.1229 | 1.0000 | 0.6193 | 0.4562 |
| | opt-66b | -0.1209 | -0.2675 | -0.2466 | 0.0646 | 0.0303 | 0.6193 | 1.0000 | 0.8655 |
| | opt-iml-max-1.3b | -0.1660 | -0.2963 | -0.3122 | 0.0440 | -0.0324 | 0.4562 | 0.8655 | 1.0000 |

Table 19: The correlations between the macro average F1-score performance of the cross-generator on the English, Spanish, and Russian languages.

| Train Language | Train LLM | text-davinci-003 | gpt-3.5-turbo | gpt-4 | alpaca-lora-30b | vicuna-13b | llama-65b | opt-66b | opt-iml-max-1.3b |
|---|---|---|---|---|---|---|---|---|---|
| en | text-davinci-003 | 0.6617 (±0.14) | 0.6860 (±0.16) | 0.6598 (±0.15) | 0.6182 (±0.11) | 0.6395 (±0.12) | 0.3917 (±0.06) | 0.4195 (±0.07) | 0.4185 (±0.10) |
| | gpt-3.5-turbo | 0.6643 (±0.14) | 0.7047 (±0.17) | 0.6891 (±0.15) | 0.6328 (±0.11) | 0.6609 (±0.12) | 0.4041 (±0.06) | 0.4083 (±0.08) | 0.3967 (±0.07) |
| | gpt-4 | 0.5993 (±0.11) | 0.6641 (±0.14) | 0.6898 (±0.16) | 0.5517 (±0.08) | 0.5993 (±0.09) | 0.3942 (±0.06) | 0.3954 (±0.06) | 0.3687 (±0.02) |
| | alpaca-lora-30b | 0.6826 (±0.13) | 0.6876 (±0.16) | 0.6679 (±0.16) | 0.6753 (±0.11) | 0.6535 (±0.12) | 0.4519 (±0.06) | 0.4416 (±0.06) | 0.4562 (±0.08) |
| | vicuna-13b | 0.6488 (±0.10) | 0.6619 (±0.15) | 0.6517 (±0.14) | 0.6143 (±0.09) | 0.6476 (±0.12) | 0.4652 (±0.05) | 0.4620 (±0.09) | 0.4563 (±0.13) |
| | llama-65b | 0.4551 (±0.08) | 0.4466 (±0.07) | 0.4754 (±0.06) | 0.4640 (±0.06) | 0.5048 (±0.07) | 0.5713 (±0.10) | 0.5391 (±0.07) | 0.5542 (±0.09) |
| | opt-66b | 0.4925 (±0.05) | 0.4989 (±0.06) | 0.5366 (±0.06) | 0.4708 (±0.02) | 0.5483 (±0.06) | 0.6285 (±0.08) | 0.5587 (±0.05) | 0.5620 (±0.08) |
| | opt-iml-max-1.3b | 0.5614 (±0.06) | 0.5460 (±0.07) | 0.5430 (±0.06) | 0.5745 (±0.07) | 0.6196 (±0.10) | 0.6617 (±0.10) | 0.6280 (±0.10) | 0.6832 (±0.12) |
| es | text-davinci-003 | 0.7718 (±0.11) | 0.7837 (±0.14) | 0.7114 (±0.16) | 0.6968 (±0.09) | 0.7294 (±0.11) | 0.4163 (±0.04) | 0.4246 (±0.04) | 0.4392 (±0.10) |
| | gpt-3.5-turbo | 0.7236 (±0.13) | 0.7791 (±0.15) | 0.7270 (±0.15) | 0.6585 (±0.10) | 0.7007 (±0.11) | 0.4144 (±0.07) | 0.3976 (±0.03) | 0.3919 (±0.06) |
| | gpt-4 | 0.6631 (±0.15) | 0.7333 (±0.16) | 0.7503 (±0.14) | 0.5998 (±0.13) | 0.6668 (±0.12) | 0.4244 (±0.05) | 0.3917 (±0.04) | 0.3795 (±0.05) |
| | alpaca-lora-30b | 0.7414 (±0.12) | 0.7398 (±0.13) | 0.6540 (±0.14) | 0.7573 (±0.11) | 0.7128 (±0.12) | 0.5235 (±0.06) | 0.4818 (±0.03) | 0.5226 (±0.08) |
| | vicuna-13b | 0.7439 (±0.11) | 0.7713 (±0.11) | 0.7161 (±0.12) | 0.6959 (±0.08) | 0.7616 (±0.09) | 0.5375 (±0.05) | 0.4754 (±0.04) | 0.5260 (±0.10) |
| | llama-65b | 0.4788 (±0.08) | 0.4753 (±0.09) | 0.4877 (±0.07) | 0.5378 (±0.06) | 0.5702 (±0.06) | 0.7655 (±0.08) | 0.6233 (±0.10) | 0.6813 (±0.13) |
| | opt-66b | 0.5669 (±0.06) | 0.5156 (±0.08) | 0.5016 (±0.09) | 0.5901 (±0.04) | 0.6245 (±0.07) | 0.6707 (±0.08) | 0.6780 (±0.10) | 0.6968 (±0.10) |
| | opt-iml-max-1.3b | 0.4749 (±0.10) | 0.4367 (±0.11) | 0.4129 (±0.10) | 0.5335 (±0.07) | 0.5480 (±0.06) | 0.6275 (±0.09) | 0.6550 (±0.10) | 0.8191 (±0.11) |
| ru | text-davinci-003 | 0.6325 (±0.13) | 0.5549 (±0.16) | 0.4914 (±0.12) | 0.6171 (±0.15) | 0.5626 (±0.12) | 0.4055 (±0.06) | 0.4656 (±0.07) | 0.4830 (±0.13) |
| | gpt-3.5-turbo | 0.6759 (±0.21) | 0.7272 (±0.21) | 0.7087 (±0.21) | 0.6492 (±0.18) | 0.6688 (±0.20) | 0.4152 (±0.03) | 0.4130 (±0.04) | 0.3937 (±0.03) |
| | gpt-4 | 0.6449 (±0.18) | 0.6905 (±0.19) | 0.7013 (±0.19) | 0.6032 (±0.13) | 0.6353 (±0.17) | 0.4238 (±0.04) | 0.4186 (±0.03) | 0.3898 (±0.05) |
| | alpaca-lora-30b | 0.6520 (±0.19) | 0.6620 (±0.16) | 0.6284 (±0.15) | 0.6968 (±0.17) | 0.6417 (±0.18) | 0.5504 (±0.06) | 0.5275 (±0.11) | 0.5903 (±0.17) |
| | vicuna-13b | 0.6099 (±0.17) | 0.6175 (±0.15) | 0.6024 (±0.14) | 0.6222 (±0.16) | 0.6490 (±0.18) | 0.5536 (±0.06) | 0.5346 (±0.10) | 0.5884 (±0.18) |
| | llama-65b | 0.4663 (±0.05) | 0.4514 (±0.03) | 0.4730 (±0.04) | 0.5068 (±0.07) | 0.5212 (±0.08) | 0.6366 (±0.10) | 0.5697 (±0.11) | 0.5841 (±0.13) |
| | opt-66b | 0.4189 (±0.05) | 0.3754 (±0.04) | 0.3769 (±0.04) | 0.4698 (±0.04) | 0.4513 (±0.03) | 0.5742 (±0.13) | 0.6328 (±0.13) | 0.6938 (±0.21) |
| | opt-iml-max-1.3b | 0.4034 (±0.09) | 0.3773 (±0.06) | 0.3643 (±0.05) | 0.4413 (±0.07) | 0.4036 (±0.06) | 0.4374 (±0.07) | 0.5101 (±0.05) | 0.6612 (±0.10) |

Table 20: The mean and standard deviation between the macro average F1-score performance of the cross-generator on the English, Spanish, and Russian languages.