# OpenReview forum: "MULTITuDE: Large-Scale Multilingual Machine-Generated Text Detection Benchmark"
_EMNLP/2023/Conference — EMNLP 2023 Main_

### Official Review · Reviewer_45v3 · 2023-08-03

**Soundness:** 4

**Excitement:**

3: Ambivalent: It has merits (e.g., it reports state-of-the-art results, the idea is nice), but there are key weaknesses (e.g., it describes incremental work), and it can significantly benefit from another round of revision. However, I won't object to accepting it if my co-reviewers champion it.

**Paper Topic And Main Contributions:**

- This paper is about multilingual machine-generated text detection
- For the same task, this paper presents a new benchmarking dataset called MULTITuDE, which comprises 74,081 texts in 11 languages. The dataset includes 7,992 human-written (news articles) texts and 66,089 machine-generated texts.
- This paper provides a comprehensive multilingual benchmark for several SOTA detection methods and also evaluate the cross-language generalization of finetuned models in multilingual setting

**Reasons To Accept:**

- Dataset: The paper introduces MULTITuDE, a novel benchmarking dataset comprising 74,081 texts in 11 languages, including both human-written and machine-generated texts.

- Comprehensive Benchmark: The paper provides a thorough multilingual benchmark for SOTA detection methods, enhancing the understanding of existing techniques.

- The related works section is well-written and thoroughly covers relevant literature in the field.


**Reasons To Reject:**

- While the paper makes contributions in terms of the dataset and evaluation, its novelty beyond these aspects might be relatively limited

**Reproducibility:**

5: Could easily reproduce the results.

**Reviewer Confidence:**

3: Pretty sure, but there's a chance I missed something. Although I have a good feel for this area in general, I did not carefully check the paper's details, e.g., the math, experimental design, or novelty.

---

> ### Author Rebuttal · Authors · 2023-08-25
>
> We thank the reviewer for the provided review, comments, and suggestions for improvements of our paper.
>
> > While the paper makes contributions in terms of the dataset and evaluation, its novelty beyond these aspects might be relatively limited
>
> The contributed benchmark reveals existing detectors’ capability of cross-lingual usage (usage on unseen languages). Based on our results, one can predict on which unseen languages a detector could perform better than the others (linguistic similarity, script). Until now, these intuitions were just in the form of assumptions.
>
> Even more, our study also provided some unexpected results, such as the English is a sub-optimal default language for cross-lingual usage (i.e. Spanish-trained detector outperformed English-trained even on languages linguistically more similar to English), although English is widely used for many NLP tasks for monolingual training to be used in a cross-lingual setting (this should be changed at least for MGTD). We have also found out that script matters a lot, since we can see different behavior for Latin and non-Latin script languages (see Czech results in Tab. 4, and correlation of non-Latin script languages in Tab. 5).
>
> In addition, the paper introduces a novel experimental design. Thanks to a source code, which will be openly published on Github, researchers can build on the top of our results and extend experiments with investigating additional languages, content types (e.g., social media posts, reviews), text generators or detectors, that will appear in future.
>
> Finally, the generated dataset itself (by including texts generated by a wide range of SOTA LLMs with multilingual capabilities) provides additional novel opportunities to be utilized also for tasks beyond the detection of the machine generated text and authorship attribution, such as obtaining insights for low-resource generators/detectors, analysis of characteristics of generated multilingual texts or code-switching neural text detection. In the Discussion and Conclusion sections of the revised paper, we will signify the unexpected results and additional novelties of our outcomes.

---

### Official Review · Reviewer_Z4xe · 2023-08-04

**Typos Grammar Style And Presentation Improvements:** N/A
**Soundness:** 4

**Excitement:**

3: Ambivalent: It has merits (e.g., it reports state-of-the-art results, the idea is nice), but there are key weaknesses (e.g., it describes incremental work), and it can significantly benefit from another round of revision. However, I won't object to accepting it if my co-reviewers champion it.

**Missing References:**

N/A

**Paper Topic And Main Contributions:**

The paper presents a large-scale evaluation of machine-generated text detection techniques for 11 languages and using 11 LLMs.
The choice of languages and techniques was carefully done to allow studying certain phenomena, such as linguistic similarity.


**Questions For The Authors:**

How come the study was limited to these 11 languages?
What do you anticipate to be additional challenges when working with other langauges?

**Reasons To Accept:**

* large scale experiment
* good selections of experimental variables.
* useful insights.
* well written.

**Reasons To Reject:**

* No statistical significance results are presenteed.

**Reproducibility:**

4: Could mostly reproduce the results, but there may be some variation because of sample variance or minor variations in their interpretation of the protocol or method.

**Reviewer Confidence:**

4: Quite sure. I tried to check the important points carefully. It's unlikely, though conceivable, that I missed something that should affect my ratings.

---

> ### Author Rebuttal · Authors · 2023-08-25
>
> We thank the reviewer for the provided review, comments, and suggestions for improvements of our paper.
>
> > No statistical significance results are presenteed.
>
> Based on the reviewer’s comment, we have tested the statistical significance of the results presented in the paper which we will add into the revised version of the paper.
>
> Firstly, **we tested for each test language whether the observed differences between different train languages/setups (en, es, ru, all, en3) as reported in Tab. 4 are statistically significant**. To do this, we have conducted repeated measures ANOVA tests for each test language (having F1-score for a given test language as a dependent variable, the combinations of detectors and text generators as “subjects” and train language/setup as an independent within-subjects variable). For all 11 test languages, the observed differences are statistically significant (p < 0.05). Next, we have conducted post-hoc pairwise tests between pairs of train languages/setups per each test language for a more in depth analysis.
>
> The results of these tests show that using all 3 train languages combined (`all` in Tab. 4) for fine-tuning has statistically significantly better performance in 4 cases (cs, de, nl, pt). In the case of the 3 languages for which we also have train data (en, es, ru), the differences between `all` and using just the train language itself are not statistically significant. Similarly, in the case of languages which are very similar to one of the train languages (ca - es, uk - ru), the difference between `all` and when using just the similar train language is not significant. Lastly, for the Arabic and Chinese languages, the difference between using `all` and just `ru` for training is not significant, likely showing the impact of a different script in these languages which confirms the claims in the paper.
>
> Overall, we can conclude that multilingual fine-tuning is better than monolingual one as it can generalize to a wider range of unseen languages and shows overall the best performance, but the language similarity and differences in script play an important role and monolingual fine-tuning can have (for linguistically similar languages) results which are on par with the multilingual fine-tuning.
>
> We will include the results of these tests into the paper (together with computed confidence intervals for the reported mean F1 scores) and extend the discussion accordingly.
>
> Additionally, we **tested statistical significance of the correlations presented in the correlation matrix in Tab. 5 as well as calculated the 95% confidence intervals**. All correlations presented in the table (except for pairs with little or no correlation observed) are statistically significant, thus supporting our conclusions of language and script similarity. The results will be added in the revised paper for Tab. 5, and analogously will be executed for Tab. 6.
>
>
> >  How come the study was limited to these 11 languages? What do you anticipate to be additional challenges when working with other langauges?
>
> We understand the reviewer’s point, and can confirm that more languages could be added if we have more resources and time for empirical validation (for instance, 64 languages in MassiveSumm include >300 samples even after our preprocessing steps, i.e. could be used in MULTITuDE test set). The number of available human-text samples was not a problem, even for including more languages in the train split (58 languages contained >1k samples). However, we needed to take the dimensionality of experiments into account, considering combinations of train languages and text-generation LLMs (machine texts sources). Even now, there were 45 fine-tuned versions of each detector base model (i.e., 315 fine-tuned detection models). Therefore, each additional train language represents additional 63 fine-tuned detection models.
>
> Given such a vast number, we needed to consider the computational feasibility and pick appropriate combinations of the representatives. The languages selected for training and testing were carefully selected to enable evaluation of cross-lingual generalization to similar and dissimilar languages. Two test languages related to each train language were just enough to show a trend, two test languages completely unrelated to train languages were also enough to evaluate such a generalization effect. Major limitations of working with all available languages are costs and time resources related to machine-text generation for each language, fine-tuning combinations, along with evaluation of commercial detectors on all test languages.
>
> We will update Sec. 3 about selecting only representatives to see the effect and answer our research questions and avoid waste of resources (including more languages than necessary).
>
> Since we will release full source code used in dataset creation, the researchers will easily be able to extend data to get more human-text samples (for already included languages as well as for other languages). Also for getting machine-generated texts, the released scripts can be used.
>
> > Reproducibility: 4: Could mostly reproduce the results, but there may be some variation because of sample variance or minor variations in their interpretation of the protocol or method.
>
> All source code, including dataset creation, detectors fine-tuning, detectors usage, and evaluation, will be available on GitHub. We have used fixed random seed values where possible, thus reproducibility should be completely possible.

---

### Official Review · Reviewer_tckt · 2023-08-05

**Soundness:** 3

**Excitement:**

3: Ambivalent: It has merits (e.g., it reports state-of-the-art results, the idea is nice), but there are key weaknesses (e.g., it describes incremental work), and it can significantly benefit from another round of revision. However, I won't object to accepting it if my co-reviewers champion it.

**Paper Topic And Main Contributions:**

The paper proposes a multilingual corpus of machine-generated texts (MGT) with the aim to facilitate MGT detection. The main difference from the previous work is that the authors leverage novel generators like ChatGPT and provide the data for 11 languages, though only 3 of them (Russian, English, and Spanish) have training data. The human-written data is quite small: 8,000 texts for all languages both for training and testing. A notable drawback of the developed corpus is that it encompasses only one domain, in particular news texts.

The main contribution of the work is empirical. The authors conduct a vast empirical investigation in various settings: cross-lingual, analyzing generalizability across generators, etc. However, all findings are quite straightforward: (1) for better generalization the languages and generators should be similar; (2) training on more diverse multilingual data helps to improve cross-lingual performance compared to training on small monolingual data, which is expected, and is known from previous work that conducts cross-lingual experiments.

The experimental investigation is vast but lacks several important things. Authors fine-tune a battery of randomly chosen models from HuggingFace but omit hyperparameter optimization.
The authors note that they find a single set of hyperparameters that demonstrate good performance for all considered models. However, these models significantly diverge in architecture and size, therefore, one set of hyperparameters cannot provide an optimal performance for all of them. They also use default hyperparameters in some cases (random forest). The results obtained with nonoptimal hyperparameters could be unreliable and invalid for supporting the claims made in the paper.


**Questions For The Authors:**

-	Why you do not provide more training data for most languages. E.g. for Ukrainian and German you could use Wikipedia as a source of human-written texts.
-	Could you clarify the claim made on lines: 480-485. Why presence of English in the training data results in weak detection performance?
-	Why in Table 3 mGPT is trained only on ru and ELECTRA-large trained only on en?


**Reasons To Accept:**

-	The paper provides another novel corpus with automatically generated texts. 3 languages have some training data and for 8 there is test data.
-	Vast empirical investigation in a cross-lingual setting.


**Reasons To Reject:**

-	The constructed resource encompasses only one domain: news texts.
-	There is a limited amount of training data only for 3 languages; also a limited amount of human-written texts.
-	Despite experiments are vast the findings are quite straightforward: (1) similarity between generators and languages helps; (2) more diverse data in multiple languages helps generalization.
-	The quality of experiments is questionable due to lack of hyperparameter optimization.


**Reproducibility:**

3: Could reproduce the results with some difficulty. The settings of parameters are underspecified or subjectively determined; the training/evaluation data are not widely available.

**Reviewer Confidence:**

4: Quite sure. I tried to check the important points carefully. It's unlikely, though conceivable, that I missed something that should affect my ratings.

**Typos Grammar Style And Presentation Improvements:**

Line 437: XML-RoBERTa => XLM-RoBERTa
Line 302: remains ambiguous => remains unknown ?
Line 297: popularly used => popular
Line 140: it is hard to agree with the claim that LMs became ubiquitous in 2018. LMs have been used long before that.

---

> ### Author Rebuttal · Authors · 2023-08-25
>
> We thank the reviewer for the provided review, comments, and suggestions for improvements of our paper.
>
> > The experimental investigation is vast but lacks several important things. Authors fine-tune a battery of randomly chosen models from HuggingFace (...)
>
> Please note that the models used as detector base models in our paper were not randomly chosen, but rather very carefully chosen, with respect to how state-of-the-art it is and the experimentation resources that we have. We have primarily used multilingual base models (XLM-RoBERTa, BERT-multilingual, mGPT, and MDeBERTa) that are publicly available and belong to the SOTA multilingual pre-trained models used for a wide range of downstream tasks. Besides these, we have also used English-only pretrained models that have been commonly used as detectors in previous studies, see e.g. [1, 2] below. We used these to see how they would perform on non-English language datasets. In this group, there were RoBERTa-large-OpenAI-detector, GPT2, and ELECTRA (already pre-trained as a discriminator).
>
> We will add the detailed reasoning behind the base-models selection to the Appendix C.
>
> [1] Uchendu, A., Ma, Z., Le, T., Zhang, R., & Lee, D. (2021). Turingbench: A benchmark environment for turing test in the age of neural text generation. EMNLP'20
>
> [2] Zhong, W., Tang, D., Xu, Z., Wang, R., Duan, N., Zhou, M., ... & Yin, J. (2020). Neural deepfake detection with factual structure of text. EMNLP'20
>
>
> > The constructed resource encompasses only one domain: news texts.
>
> We agree that it would be interesting to show what are the results for another domain (e.g., social media posts, discussion posts, reviews), and provide cross-domain evaluation. It is one of our next steps in this research. At the time of research design, across many choices, the team made a conscious decision to focus first on the multi-linguality of AI-generated texts over multi-domain to decrease dimensionality of the problem and so that the results would not be biased to domain specifics instead of languages. Each new domain would multiply the amount of fine-tuned detection methods and as the result, the magnitude of the results would be simply too high to include in a single paper. However, in the future work, we plan to examine additional domains (especially social media posts).
>
> > The human-written data is quite small: 8,000 texts for all languages both for training and testing.  (...) There is a limited amount of training data only for 3 languages; also a limited amount of human-written texts.
>
> We have carefully selected the train and test languages to be able to evaluate cross-lingual generalization to linguistically similar and dissimilar languages. For each train language, there are two or more test languages belonging to the same language family branch (e.g., for English as the train language, there are Dutch and German as the related test languages). In addition, there are two more languages in the test set, which are completely unrelated to the train languages (Arabic and Chinese). This information has already been included in Sec. 3 in the submitted paper.
>
> Regarding the chosen numbers of samples, the information about our reasoning was indeed omitted in the submitted paper due to page limit. We will add more information about the reasoning about the language selection and the dataset size in Appendix B.1. In our preliminary study, we have extensively experimented (using TuringBench English data, AuTexTification Spanish data, and RuATD Russian data) and chose a minimal number of samples needed to fine-tune the detectors and properly evaluate them, where we compared the performances using the selected smaller amount of samples (500, 600, 1000, 1500 human samples where available) and all samples available (5000 for English, 1150 for Spanish, 2450 for Russian). These experiments resulted in 1k human samples and 1k samples per text-generator required for training, with negligible drop in the performance of fine-tuned detectors (i.e., within 5%). Considering our broad spectrum of experiments, we needed to minimize the amount of data to reduce GPU-hours (environmental impact), costs, and time, and thus chose the said numbers in the paper.
>
> In addition, since we are experimenting with smaller train sets (regarding the number of samples), we have provided 3k human texts for English (for “en3” version of train language in Tab. 4) to ensure that the performance effect is not simply due to a higher number of train samples in case of multilingually fine-tuned detectors (“all” version of train language in Tab. 4).
>
> > Despite experiments are vast the findings are quite straightforward: (1) similarity between generators and languages helps; (2) more diverse data in multiple languages helps generalization.
>
> Our reported findings (which the reviewer has cited) may seem straightforward and intuitive to experts, but please note that these findings have not been convincingly confirmed by experiments in the existing studies, especially regarding machine-generated text detection (i.e., not evaluated on this task before). Until now, experts have suspected those intuitions to hold but never corroborated with convincing evidence. There are differences even among different binary text-classification tasks regarding cross-lingual generalization (e.g., hate-speech detection, machine-generated text detection); therefore, such effects must be thoroughly investigated. For example, Artetxe et al. [3] showed slightly different cross-lingual capability of models for various tasks and indicates that multilingual training is not necessary (in contrast to our results showing that it is important for the MGTD task). Hu et al. [4] also confirmed that a model’s cross-lingual capability varies significantly both between tasks and languages.
>
> Even more, **our study also provided some unexpected results**, such as the English is a sub-optimal default language for cross-lingual usage (i.e. Spanish-trained detector outperformed English-trained), although English is widely used for many NLP tasks for monolingual training to be used in cross-lingual setting (this should be changed at least for MGTD). We have also found out that script matters a lot, since we can see different behavior for Latin and non-Latin script languages (see Czech results in Tab. 4, and correlation of non-Latin script languages in Tab. 5).
>
> [3] Artetxe, M., Ruder, S., & Yogatama, D. (2019). On the cross-lingual transferability of monolingual representations. ACL'20. https://arxiv.org/abs/1910.11856
>
> [4] Hu, J., Ruder, S., Siddhant, A., Neubig, G., Firat, O., & Johnson, M. (2020). XTREME: A massively multilingual multi-task benchmark for evaluating cross-lingual generalisation. In International Conference on Machine Learning (pp. 4411-4421). PMLR.
>
> > The results obtained with nonoptimal hyperparameters could be unreliable and invalid for supporting the claims made in the paper. (...) The quality of experiments is questionable due to lack of hyperparameter optimization.
>
> Hyperparameters are optimized in a unified way common for all detector base models to cope with the complexity of such a task (considering 315 fine-tuned models). It was our intention to optimize hyperparameters in a unified way for all fine-tuned detection models, *NOT* to come up with the best MGT detector but to study the main trends and patterns among baseline cross-lingual generalization of monolingually and multilingually fine-tuned and off-the-shelf detectors. It is true that such an approach might not result in the best performance for all detector base models (could affect Tab. 3), but it does *NOT* affect experiments that answer our research questions (cross-lingual and cross-generator generalization). Using adaptive learning rate and early stopping mechanism should serve sufficiently to cope with various architecture sizes to train detectors.
>
> We did not originally optimize hyperparameters for statistical detectors. As mentioned in Sec. 5.1, we used the MGTBench implementation (https://github.com/xinleihe/MGTBench) that was used previously in other recent works [5, 6] as a baseline. This implementation uses default parameters for Logistic Regression.
>
> **Based on the reviewer’s feedback, we have recomputed results for our modification of the entropy-based statistical detector using Random Forest for binary classification.** The results with hyperparameters optimized using Randomized Grid Search with 5-fold cross-validation and 1k of iterations are very similar to the ones in the paper. The following line in Tab. 3:
>
>     Entropy + RandomForest* S N/A N/A 0.4811 0.8347 0.8040 0.8781 0.8781 0.9845 0.0148
>
> will be changed to:
>
>     Entropy + RandomForest* S N/A N/A 0.4863 0.8335 0.8050 0.8729 0.8729 0.9756 0.0217
>
> Also, the following line in Tab. 10:
>
>     Entropy + RandomForest 0.4800 0.4689 0.4701 0.4735 0.4705 0.4704 0.4696 0.4706 0.5020 0.5011 0.4682
>
> will be changed to:
>
>     Entropy + RandomForest 0.4860 0.4721 0.4732 0.4729 0.4703 0.4697 0.4692 0.4702 0.5202 0.5040 0.4663
>
> We will update the results in the revised version of the paper accordingly. We will also add details regarding hyperparameter optimization in Sec. 5.1 in the revised paper. The grid consisted from the following parameters:
>
> + n_estimators = [10, 50, 100, 150, 300] # number of trees in the random forest
>
> + criterion = ['gini', 'entropy'] # function to measure the quality of a split
>
> + max_features = ['sqrt', 'log2', None] # number of features in consideration at every split
>
> + max_depth = [None, 10, 100] # maximum number of levels allowed in each decision tree
>
> + min_samples_split = [2, 4, 6] # minimum sample number to split a node
>
> + min_samples_leaf = [1, 3] # minimum sample number that can be stored in a leaf node
>
> + bootstrap = [True, False] # method used to sample data points
>
> [5] Yu, X., Qi, Y., Chen, K., Chen, G., Yang, X., Zhu, P., ... & Yu, N. (2023). GPT Paternity Test: GPT Generated Text Detection with GPT Genetic Inheritance. arXiv preprint arXiv:2305.12519.
>
> [6] Wu, Z., & Xiang, H. (2023). MFD: Multi-Feature Detection of LLM-Generated Text.
>
> > Why you do not provide more training data for most languages. E.g. for Ukrainian and German you could use Wikipedia as a source of human-written texts.
>
> We understand the reviewer’s point, however, the number of available human texts for other languages was not a problem in our data source. We have focused strictly on the news domain (to reduce dimensionality of the problem) and intentionally used a common source of human texts for our dataset (human texts from MassiveSumm dataset). There are a number of languages with higher amounts of samples available in the MassiveSumm dataset (even after our preprocessing steps, 58 languages containing over 1k human texts - i.e., usable in our train set, 64 languages containing over 300 human texts - i.e., usable in our test set). However, we needed to take dimensionality into account, considering combinations of train languages and text-generation LLMs (machine texts sources). Even now, there were 45 fine-tuned versions of each detector base model (i.e., 315 fine-tuned detection models). Therefore, each additional train language represents additional 63 fine-tuned detection models. Given such a vast number, we needed to consider the computational feasibility and pick appropriate representatives. The languages selected for training and testing were carefully selected to enable evaluation of cross-lingual generalization to similar and dissimilar languages.
>
> > Could you clarify the claim made on lines: 480-485. Why presence of English in the training data results in weak detection performance?
>
> 480-485: “We hypothesize that this is caused by the fact that English is often the most common language in the training data for both the generators and the detectors, which might lead to different behavior for this particular language (e.g., the perplexity might be lower).”
>
> Please note that it is rather a hypothesis (an intuition). Maybe there is a confusion about what we meant by “training data” in that sentence (it was not our training data used for fine-tuning, but training data used in pretraining of LLMs - text generators as well as detector base models). Fine-tuning on English does not decrease detection performance on English, but provides worse generalization to other languages than fine-tuning on Spanish or Russian. Our hypothesis to explain this behavior is that detectors behave differently on English and non-English data due to its predominance in detector base models pretraining and also in pretraining of text generation LLMs. We will clarify this in the revised paper (“training data” → “pre-training data”, “different behavior” → “different behavior (regarding cross-lingual capability)”). This hypothesis needs to be experimented and corroborated in our future study.
>
> > Why in Table 3 mGPT is trained only on ru and ELECTRA-large trained only on en?
>
> Since there are 315 fine-tuned detection methods (45 versions of each detector base model) which could not be all provided in the table in the paper (will be available on GitHub), Tab. 3 contains only the best performing (based on Macro average of F1-score) version for each detector base model. For those two detectors the reviewer mentioned, the performance was higher (although not by a large margin) for a version trained on a single language rather than on a combination of languages.
> This information is provided in lines 356-364; however, since it is not clear, we will also add a note regarding the number of 315 fine-tuned detection methods and that it would be impossible to show all of them in a single table in the paper. In the revised paper, we will also add information about showing only the best-performing version of individual detector base models directly in the Tab. 3 caption.
>
> > Typos Grammar Style And Presentation Improvements: (…)
>
> We will address all the identified typos in the revised paper and screen the paper for all we can find.
>
> > Could reproduce the results with some difficulty. The settings of parameters are underspecified or subjectively determined; the training/evaluation data are not widely available.
>
> All source code, including dataset creation, detectors fine-tuning, detectors usage, and evaluation, will be available on GitHub. We have used fixed random seed values where possible, thus reproducibility should be completely possible.

---

### Meta-Review · Area_Chair_GeDe · 2023-09-19

**Recommendation:** 4

**Metareview:**

This paper presents a novel large-scale benchmark dataset for multilingual machine-generated text (MGT) detection, named MULTITuDE. The dataset is comprised of 74,081 texts in 11 languages and includes both human-written and machine-generated texts. To explore the performance of current state-of-the-art detection methods, the authors conduct a comprehensive multilingual benchmark. They also evaluate the cross-language generalization capabilities of fine-tuned models in a multilingual setting.

Main Contributions:

The authors introduce MULTITuDE, a new benchmarking dataset for multilingual machine-generated text detection, which includes 7,992 human-written texts and 66,089 machine-generated texts.

The paper provides a comprehensive multilingual benchmark for state-of-the-art detection methods, offering a detailed study of these techniques across 11 languages and employing 11 LLMs.

The authors evaluate the cross-language generalization capabilities of fine-tuned models in a multilingual setting, offering valuable insights into the performance and adaptability of these models.

Reasons for Acceptance:

The introduction of the MULTITuDE dataset provides a valuable resource for the research community, fostering further study in multilingual machine-generated text detection.

The comprehensive benchmarking and evaluation of state-of-the-art detection methods enhance understanding of these techniques and their performance across different languages.

The authors' exploration of cross-language generalization provides useful insights into the performance of fine-tuned models in multilingual settings.

Reasons for Rejection:

The constructed resource is limited to one domain (news texts), which may limit its applicability to other domains.

There is a limited amount of training data available for only three languages, and the amount of human-written text is also limited.

The experimental findings, while extensive, are quite straightforward and expected.

The quality of the experiments may be questionable due to the lack of hyperparameter optimization, which could affect the reliability of the results and the validity of the claims made in the paper.

---

### Decision · Program_Chairs · 2023-10-07

**Decision:**

Accept-Main

**Comment:**

This paper presents a novel large-scale benchmark dataset for multilingual machine-generated text (MGT) detection, named MULTITuDE. The dataset is comprised of 74,081 texts in 11 languages and includes both human-written and machine-generated texts. To explore the performance of current state-of-the-art detection methods, the authors conduct a comprehensive multilingual benchmark. They also evaluate the cross-language generalization capabilities of fine-tuned models in a multilingual setting.

Main Contributions:

The authors introduce MULTITuDE, a new benchmarking dataset for multilingual machine-generated text detection, which includes 7,992 human-written texts and 66,089 machine-generated texts.

The paper provides a comprehensive multilingual benchmark for state-of-the-art detection methods, offering a detailed study of these techniques across 11 languages and employing 11 LLMs.

The authors evaluate the cross-language generalization capabilities of fine-tuned models in a multilingual setting, offering valuable insights into the performance and adaptability of these models.

Reasons for Acceptance:

The introduction of the MULTITuDE dataset provides a valuable resource for the research community, fostering further study in multilingual machine-generated text detection.

The comprehensive benchmarking and evaluation of state-of-the-art detection methods enhance understanding of these techniques and their performance across different languages.

The authors' exploration of cross-language generalization provides useful insights into the performance of fine-tuned models in multilingual settings.

Reasons for Rejection:

The constructed resource is limited to one domain (news texts), which may limit its applicability to other domains.

There is a limited amount of training data available for only three languages, and the amount of human-written text is also limited.

The experimental findings, while extensive, are quite straightforward and expected.

The quality of the experiments may be questionable due to the lack of hyperparameter optimization, which could affect the reliability of the results and the validity of the claims made in the paper.